# A Generic Acceleration Framework
# for Stochastic Composite Optimization

**Andrei Kulunchakov and Julien Mairal**
Univ. Grenoble Alpes, Inria, CNRS, Grenoble INP, LJK, 38000 Grenoble, France
andrei.kulunchakov@inria.fr and julien.mairal@inria.fr

## Abstract

In this paper, we introduce various mechanisms to obtain accelerated first-order stochastic optimization algorithms when the objective function is convex or strongly convex. Specifically, we extend the Catalyst approach originally designed for deterministic objectives to the stochastic setting. Given an optimization method with mild convergence guarantees for strongly convex problems, the challenge is to accelerate convergence to a noise-dominated region, and then achieve convergence with an optimal worst-case complexity depending on the noise variance of the gradients. A side contribution of our work is also a generic analysis that can handle inexact proximal operators, providing new insights about the robustness of stochastic algorithms when the proximal operator cannot be exactly computed.

## 1 Introduction

In this paper, we consider stochastic composite optimization problems of the form

$$\min_{x \in \mathbb{R}^p} \{F(x) := f(x) + \psi(x)\} \quad \text{with} \quad f(x) = \mathbb{E}_\xi[\tilde{f}(x, \xi)], \tag{1}$$

where the function $f$ is convex, or $\mu$-strongly convex, and $L$-smooth (meaning differentiable with $L$-Lipschitz continuous gradient), and $\psi$ is a possibly non-smooth convex lower-semicontinuous function. For instance, $\psi$ may be the $\ell_1$-norm, which is known to induce sparsity, or an indicator function of a convex set [21]. The random variable $\xi$ corresponds to data samples. When the amount of training data is finite, the expectation $\mathbb{E}_\xi[\tilde{f}(x, \xi)]$ can be replaced by a finite sum, a setting that has attracted a lot of attention in machine learning recently, see, *e.g.*, [13, 14, 19, 25, 35, 42, 53] for incremental algorithms and [1, 26, 30, 33, 47, 55, 56] for accelerated variants.

Yet, as noted in [8], one is typically not interested in the minimization of the empirical risk—that is, a finite sum of functions—with high precision, but instead, one should focus on the expected risk involving the true (unknown) data distribution. When one can draw an infinite number of samples from this distribution, the true risk (1) may be minimized by using appropriate stochastic optimization techniques. Unfortunately, fast methods designed for deterministic objectives would not apply to this setting; methods based on stochastic approximations admit indeed optimal "slow" rates that are typically $O(1/\sqrt{k})$ for convex functions and $O(1/k)$ for strongly convex ones, depending on the exact assumptions made on the problem, where $k$ is the number of noisy gradient evaluations [38].

Better understanding the gap between deterministic and stochastic optimization is one goal of this paper. Specifically, we are interested in Nesterov's acceleration of gradient-based approaches [39, 40]. In a nutshell, gradient descent or its proximal variant applied to a $\mu$-strongly convex $L$-smooth function achieves an exponential convergence rate $O((1 - \mu/L)^k)$ in the worst case in function values, and a sublinear rate $O(L/k)$ if the function is simply convex ($\mu = 0$). By interleaving the algorithm with clever extrapolation steps, Nesterov showed that faster convergence could be achieved, and the previous convergence rates become $O((1 - \sqrt{\mu/L})^k)$ and $O(L/k^2)$, respectively. Whereas

no clear geometrical intuition seems to appear in the literature to explain why acceleration occurs, proof techniques to show accelerated convergence [5, 40, 50] and extensions to a large class of other gradient-based algorithms are now well established [1, 10, 33, 41, 47].

Yet, the effect of Nesterov's acceleration to stochastic objectives remains poorly understood since existing unaccelerated algorithms such as stochastic mirror descent [38] and their variants already achieve the optimal asymptotic rate. Besides, negative results also exist, showing that Nesterov's method may be unstable when the gradients are computed approximately [12, 16]. Nevertheless, several approaches such as [4, 11, 15, 17, 18, 23, 28, 29, 52] have managed to show that acceleration may be useful to forget faster the algorithm's initialization and reach a region dominated by the noise of stochastic gradients; then, "good" methods are expected to asymptotically converge with a rate exhibiting an optimal dependency in the noise variance [38], but with no dependency on the initialization. A major challenge is then to achieve the optimal rate for these two regimes.

In this paper, we consider an optimization method $\mathcal{M}$ with the following property: given an auxiliary strongly convex objective function $h$, we assume that $\mathcal{M}$ is able to produce iterates $(z_t)_{t \geq 0}$ with expected linear convergence to a noise-dominated region—that is, such that

$$\mathbb{E}[h(z_t) - h^\star] \leq C(1 - \tau)^t (h(z_0) - h^\star) + B\sigma^2, \tag{2}$$

where $C, \tau, B > 0$, $h^\star$ is the minimum function value, and $\sigma^2$ is an upper bound on the variance of stochastic gradients accessed by $\mathcal{M}$, which we assume to be uniformly bounded. Whereas such an assumption has limitations, it remains the most standard one for stochastic optimization (see [9, 43] for more realistic settings in the smooth case). The class of methods satisfying (2) is relatively large. For instance, when $h$ is $L$-smooth, the stochastic gradient descent method (SGD) with constant step size $1/L$ and iterate averaging satisfies (2) with $\tau = \mu/L$, $B = 1/L$, and $C = 1$, see [28].

**Main contribution.** In this paper, we extend the Catalyst approach [33] to general stochastic problems.[1] Under mild conditions, our approach is able to turn $\mathcal{M}$ into a converging algorithm with a worst-case expected complexity that decomposes into two parts: the first one exhibits an accelerated convergence rate in the sense of Nesterov and shows how fast one forgets the initial point; the second one corresponds to the stochastic regime and typically depends (optimally in many cases) on $\sigma^2$. Note that even though we only make assumptions about the behavior of $\mathcal{M}$ on strongly convex sub-problems (2), we also treat the case where the objective (1) is convex, but not strongly convex.

To illustrate the versatility of our approach, we consider the stochastic finite-sum problem [7, 22, 31, 54], where the objective (1) decomposes into $n$ components $\tilde{f}(x, \xi) = \frac{1}{n} \sum_{i=1}^n \tilde{f}_i(x, \xi)$ and $\xi$ is a stochastic perturbation, coming, *e.g.*, from data augmentation or noise injected during training to improve generalization or privacy (see [28, 35]). The underlying finite-sum structure may also result from clustering assumptions on the data [22], or from distributed computing [31], a setting beyond the scope of our paper. Whereas it was shown in [28] that classical variance-reduced stochastic optimization methods such as SVRG [53], SDCA [47], SAGA [13], or MISO [35], can be made robust to noise, the analysis of [28] is only able to accelerate the SVRG approach. With our acceleration technique, all of the aforementioned methods can be modified such that they find a point $\hat{x}$ satisfying $\mathbb{E}[F(\hat{x}) - F^\star] \leq \varepsilon$ with global iteration complexity, for the $\mu$-strongly convex case,

$$\tilde{O}\left( \left( n + \sqrt{n \frac{L}{\mu}} \right) \log \left( \frac{F(x_0) - F^\star}{\varepsilon} \right) + \frac{\sigma^2}{\mu \varepsilon} \right). \tag{3}$$

The term on the left is the optimal complexity for finite-sum optimization [1, 2], up to logarithmic terms in $L, \mu$ hidden in the $\tilde{O}(.)$ notation, and the term on the right is the optimal complexity for $\mu$-strongly convex stochastic objectives [17] where $\sigma^2$ is due to the perturbations $\xi$. As Catalyst [33], the price to pay compared to non-generic direct acceleration techniques [1, 28] is a logarithmic factor.

**Other contributions.** In this paper, we generalize the analysis of Catalyst [33, 44] to handle various new cases. Beyond the ability to deal with stochastic optimization problems, our approach (i) improves Catalyst by allowing sub-problems of the form (2) to be solved approximately *in expectation*, which is more realistic than the deterministic requirement made in [33] and which is also critical

for stochastic optimization, (ii) leads to a new accelerated stochastic gradient descent algorithms for composite optimization with similar guarantees as [17, 18, 28], (iii) handles the analysis of accelerated proximal gradient descent methods with inexact computation of proximal operators, improving the results of [45] while also treating the stochastic setting.

Finally, we note that the extension of Catalyst we propose is easy to implement. The original Catalyst method introduced in [32] indeed required solving a sequence of sub-problems while controlling carefully the convergence, *e.g.*, with duality gaps. For this reason, Catalyst has sometimes been seen as theoretically appealing but not practical enough [46]. Here, we focus on a simpler and more practical variant presented later in [33], which consists of solving sub-problems with a fixed computational budget, thus removing the need to define stopping criterions for sub-problems. The code used for our experiments is available here: `http://github.com/KuluAndrej/NIPS-2019-code`.

## 2 Related Work on Inexact and Stochastic Proximal Point Methods.

Catalyst is based on the inexact accelerated proximal point algorithm [20], which consists in solving approximately a sequence of sub-problems and updating two sequences $(x_k)_{k \geq 0}$ and $(y_k)_{k \geq 0}$ by

$$x_k \approx \underset{x \in \mathbb{R}^p}{\operatorname{argmin}} \left\{ h_k(x) := F(x) + \frac{\kappa}{2} \|x - y_{k-1}\|^2 \right\} \quad \text{and} \quad y_k = x_k + \beta_k(x_k - x_{k-1}), \quad (4)$$

where $\beta_k$ in $(0, 1)$ is obtained from Nesterov's acceleration principles [40], $\kappa$ is a well chosen regularization parameter, and $\| \cdot \|^2$ is the Euclidean norm. The method $\mathcal{M}$ is used to obtain an approximate minimizer of $h_k$; when $\mathcal{M}$ converges linearly, it may be shown that the resulting algorithm (4) enjoys a better worst-case complexity than if $\mathcal{M}$ was used directly on $f$, see [33].

Since asymptotic linear convergence is out of reach when $f$ is a stochastic objective, a classical strategy consists in replacing $F(x)$ in (4) by a finite-sum approximation obtained by random sampling, leading to deterministic sub-problems. Typically without Nesterov's acceleration (with $y_k = x_k$), this strategy is often called the stochastic proximal point method [3, 6, 27, 48, 49]. The point of view we adopt in this paper is different and is based on the minimization of surrogate functions $h_k$ related to (4), but which are more general and may take other forms than $F(x) + \frac{\kappa}{2} \|x - y_{k-1}\|^2$.

## 3 Preliminaries: Basic Multi-Stage Schemes

In this section, we present two simple multi-stage mechanisms to improve the worst-case complexities of stochastic optimization methods, before introducing acceleration principles.

**Basic restart with mini-batching or decaying step sizes.** Consider an optimization method $\mathcal{M}$ with convergence rate (2) and assume that there exists a hyper-parameter to control a trade-off between the bias $B\sigma^2$ and the computational complexity. Specifically, we assume that the bias can be reduced by an arbitrary factor $\eta < 1$, while paying a factor $1/\eta$ in terms of complexity per iteration (or $\tau$ may be reduced by a factor $\eta$, thus slowing down convergence). This may occur in two cases:

- by using a mini-batch of size $1/\eta$ to sample gradients, which replaces $\sigma^2$ by $\eta\sigma^2$;
- or the method uses a step size proportional to $\eta$ that can be chosen arbitrarily small.

For instance, stochastic gradient descent with constant step size and iterate averaging is compatible with both scenarios [28]. Then, consider a target accuracy $\varepsilon$ and define the sequences $\eta_k = 1/2^k$ and $\varepsilon_k = 2B\sigma^2\eta_k$ for $k \geq 0$. We may now solve successively the problem up to accuracy $\varepsilon_k$—*e.g.*, with a constant number $O(1/\tau)$ steps of $\mathcal{M}$ when using mini-batches of size $1/\eta_k = 2^k$ to reduce the bias—and by using the solution of iteration $k$–1 as a warm restart. As shown in Appendix B, the scheme converges and the worst-case complexity to achieve the accuracy $\varepsilon$ in expectation is

$$O\left( \frac{1}{\tau} \log\left( \frac{C(F(x_0) - F^\star)}{\varepsilon} \right) + \frac{B\sigma^2 \log(2C)}{\tau\varepsilon} \right). \quad (5)$$

For instance, one may run SGD with constant step size $\eta_k/L$ at stage $k$ with iterate averaging as in [28], which yields $B = 1/L$, $C = 1$, and $\tau = \mu/L$. Then, the left term is the classical complexity $O((L/\mu) \log(1/\varepsilon))$ of the (unaccelerated) gradient descent algorithm for deterministic objectives, whereas the right term is the optimal complexity for stochastic optimization in $O(\sigma^2/\mu\varepsilon)$. Similar restart principles appear for instance in [4] in the design of a multistage accelerated SGD algorithm.

**Restart: from sub-linear to linear rate with strong convexity.** A natural question is whether asking for a linear rate in (2) for strongly convex problems is a strong requirement. Here, we show that a sublinear rate is in fact sufficient for our needs by generalizing a restart technique introduced in [18] for stochastic optimization, which was previously used for deterministic objectives in [24].

Specifically, consider an optimization method $\mathcal{M}$ such that the convergence rate (2) is replaced by

$$\mathbb{E}[h(z_t) - h^\star] \leq \frac{D\|z_0 - z^\star\|^2}{2t^d} + \frac{B\sigma^2}{2}, \tag{6}$$

where $D, d > 0$ and $z^\star$ is a minimizer of $h$. Assume now that $h$ is $\mu$-strongly convex with $D \geq \mu$ and consider restarting $s$ times the method $\mathcal{M}$, each time running $\mathcal{M}$ for constant $t' = \lceil (2D/\mu)^{1/d} \rceil$ iterations. Then, it may be shown (see Appendix B) that the relation (2) holds with constant $t = st'$, $\tau = \frac{1}{2t'}$, and $C = 1$. If a mini-batch or step size mechanism is available, we may then proceed as before and obtain a converging scheme with complexity (5), *e.g.*, by using mini-batches of exponentially increasing sizes once the method reaches a noise-dominated region, and by using a restart frequency of order $O(1/\tau)$.

# 4 Generic Multi-Stage Approaches with Acceleration

We are now in shape to introduce a generic acceleration framework that generalizes (4). Specifically, given some point $y_{k-1}$ at iteration $k$, we consider a surrogate function $h_k$ related to a parameter $\kappa > 0$, an approximation error $\delta_k \geq 0$, and an optimization method $\mathcal{M}$ that satisfy the following properties:

($\mathcal{H}_1$) $h_k$ is $(\kappa + \mu)$-strongly convex, where $\mu$ is the strong convexity parameter of $f$;

($\mathcal{H}_2$) $\mathbb{E}[h_k(x)|\mathcal{F}_{k-1}] \leq F(x) + \frac{\kappa}{2}\|x - y_{k-1}\|^2$ for $x = \alpha_{k-1}x^\star + (1 - \alpha_{k-1})x_{k-1}$, which is deterministic given the past information $\mathcal{F}_{k-1}$ up to iteration $k$–1 and $\alpha_{k-1}$ is given in Alg. 1;

($\mathcal{H}_3$) $\mathcal{M}$ can provide the exact minimizer $x_k^\star$ of $h_k$ and a point $x_k$ (possibly equal to $x_k^\star$) such that $\mathbb{E}[F(x_k)] \leq \mathbb{E}[h_k^\star] + \delta_k$ where $h_k^\star = \min_x h_k(x)$.

The generic acceleration framework is presented in Algorithm 1. Note that the conditions on $h_k$ bear similarities with estimate sequences introduced by Nesterov [40]; indeed, ($\mathcal{H}_3$) is a direct generalization of (2.2.2) from [40] and ($\mathcal{H}_2$) resembles (2.2.1). However, the choices of $h_k$ and the proof technique are significantly different, as we will see with various examples below. We also assume at the moment that the exact minimizer $x_k^\star$ of $h_k$ is available, which differs from the Catalyst framework [33]; the case with approximate minimization will be presented in Section 4.1.

---

**Algorithm 1** Generic Acceleration Framework with Exact Minimization of $h_k$

---

1: **Input:** $x_0$ (initial estimate); $\mathcal{M}$ (optimization method); $\mu$ (strong convexity constant); $\kappa$ (parameter for $h_k$); $K$ (number of iterations); $(\delta_k)_{k\geq 0}$ (approximation errors);
2: **Initialization:** $y_0 = x_0$; $q = \frac{\mu}{\mu + \kappa}$; $\alpha_0 = 1$ if $\mu = 0$ or $\alpha_0 = \sqrt{q}$ if $\mu \neq 0$;
3: **for** $k = 1, \ldots, K$ **do**
4:    Consider a surrogate $h_k$ satisfying ($\mathcal{H}_1$), ($\mathcal{H}_2$) and obtain $x_k, x_k^\star$ using $\mathcal{M}$ satisfying ($\mathcal{H}_3$);
5:    Compute $\alpha_k$ in $(0, 1)$ by solving the equation $\alpha_k^2 = (1 - \alpha_k)\alpha_{k-1}^2 + q\alpha_k$.
6:    Update the extrapolated sequence

$$y_k = x_k^\star + \beta_k(x_k^\star - x_{k-1}) + \frac{(\kappa + \mu)(1 - \alpha_k)}{\kappa}(x_k - x_k^\star) \quad \text{with} \quad \beta_k = \frac{\alpha_{k-1}(1 - \alpha_{k-1})}{\alpha_{k-1}^2 + \alpha_k}. \tag{7}$$

7: **end for**
8: **Output:** $x_k$ (final estimate).

---

**Proposition 1** (Convergence analysis for Algorithm 1). *Consider Algorithm 1. Then,*

$$\mathbb{E}[F(x_k) - F^\star] \leq \begin{cases} (1 - \sqrt{q})^k \left( 2(F(x_0) - F^\star) + \sum_{j=1}^{k}(1 - \sqrt{q})^{-j}\delta_j \right) & \text{if } \mu \neq 0 \\ \frac{2}{(k+1)^2} \left( \kappa\|x_0 - x^\star\|^2 + \sum_{j=1}^{k}\delta_j(j + 1)^2 \right) & \text{otherwise} \end{cases}. \tag{8}$$

The proof of the proposition is given in Appendix C and is based on an extension of the analysis of Catalyst [33]. Next, we present various application cases leading to algorithms with acceleration.

**Accelerated proximal gradient method.** When $f$ is deterministic and the proximal operator of $\psi$ (see Appendix A for the definition) can be computed in closed form, choose $\kappa = L - \mu$ and define

$$h_k(x) := f(y_{k-1}) + \nabla f(y_{k-1})^\top (x - y_{k-1}) + \frac{L}{2}\|x - y_{k-1}\|^2 + \psi(x). \tag{9}$$

Consider $\mathcal{M}$ that minimizes $h_k$ in closed form: $x_k = x_k^\star = \mathrm{Prox}_{\psi/L}\left[y_{k-1} - \frac{1}{L}\nabla f(y_{k-1})\right]$. Then, $(\mathcal{H}_1)$ is obvious; $(\mathcal{H}_2)$ holds from the convexity of $f$, and $(\mathcal{H}_3)$ with $\delta_k = 0$ follows from classical inequalities for $L$-smooth functions [40]. Finally, we recover accelerated convergence rates [5, 40].

**Accelerated proximal point algorithm.** We consider $h_k$ given in (4) with exact minimization (thus an unrealistic setting, but conceptually interesting) with $\kappa = L - \mu$. Then, the assumptions $(\mathcal{H}_1)$, $(\mathcal{H}_2)$, and $(\mathcal{H}_3)$ are satisfied with $\delta_k = 0$ and we recover the accelerated rates of [20].

**Accelerated stochastic gradient descent with prox.** A more interesting choice of surrogate is

$$h_k(x) := f(y_{k-1}) + g_k^\top (x - y_{k-1}) + \frac{\kappa + \mu}{2}\|x - y_{k-1}\|^2 + \psi(x), \tag{10}$$

where $\kappa \geq L - \mu$ and $g_k$ is an unbiased estimate of $\nabla f(y_{k-1})$—that is, $\mathbb{E}[g_k|\mathcal{F}_{k-1}] = \nabla f(y_{k-1})$—with variance bounded by $\sigma^2$, following classical assumptions from the stochastic optimization literature [17, 18, 23]. Then, $(\mathcal{H}_1)$ and $(\mathcal{H}_2)$ are satisfied given that $f$ is convex. To characterize $(\mathcal{H}_3)$, consider $\mathcal{M}$ that minimizes $h_k$ in closed form: $x_k = x_k^\star = \mathrm{Prox}_{\psi/(\kappa+\mu)}[y_{k-1} - \frac{1}{\kappa+\mu}g_k]$, and define $u_{k-1} := \mathrm{Prox}_{\psi/(\kappa+\mu)}[y_{k-1} - \frac{1}{\kappa+\mu}\nabla f(y_{k-1})]$, which is deterministic given $\mathcal{F}_{k-1}$. Then, from (10),

$$
\begin{aligned}
F(x_k) &\leq h_k(x_k) + (\nabla f(y_{k-1}) - g_k)^\top (x_k - y_{k-1}) \quad &\text{(from $L$-smoothness of $f$)} \\
&= h_k^\star + (\nabla f(y_{k-1}) - g_k)^\top (x_k - u_{k-1}) + (\nabla f(y_{k-1}) - g_k)^\top (u_{k-1} - y_{k-1}).
\end{aligned}
$$

When taking expectations, the last term on the right disappears since $\mathbb{E}[g_k|\mathcal{F}_{k-1}] = \nabla f(y_{k-1})$:

$$
\begin{aligned}
\mathbb{E}[F(x_k)] &\leq \mathbb{E}[h_k^\star] + \mathbb{E}[\|g_k - \nabla f(y_{k-1})\|\|x_k - u_{k-1}\|] \\
&\leq \mathbb{E}[h_k^\star] + \frac{1}{\kappa+\mu}\mathbb{E}\left[\|g_k - \nabla f(y_{k-1})\|^2\right] \leq \mathbb{E}[h_k^\star] + \frac{\sigma^2}{\kappa+\mu},
\end{aligned} \tag{11}
$$

where we used the non-expansiveness of the proximal operator [37]. Therefore, $(\mathcal{H}_3)$ holds with $\delta_k = \sigma^2/(\kappa + \mu)$. The resulting algorithm is similar to [28] and offers the same guarantees. The novelty of our approach is then a unified convergence proof for the deterministic and stochastic cases.

**Corollary 2** (Complexity of proximal stochastic gradient algorithm, $\mu > 0$)**.** *Consider Algorithm 1 with $h_k$ defined in (10). When $f$ is $\mu$-strongly convex, choose $\kappa = L - \mu$. Then,*

$$\mathbb{E}[F(x_k) - F^\star] \leq \left(1 - \sqrt{\frac{\mu}{L}}\right)^k (F(x_0) - F^\star) + \frac{\sigma^2}{\sqrt{\mu L}},$$

which is of the form (2) with $\tau = \sqrt{\mu/L}$ and $B = \sigma^2/(\sqrt{\mu L})$. Interestingly, the optimal complexity $O\left(\sqrt{L/\mu}\log((F(x_0) - F^\star)/\varepsilon) + \sigma^2/\mu\varepsilon\right)$ can be obtained by using the first restart strategy presented in Section 3, see Eq. (5), either by using increasing mini-batches or decreasing step sizes.

When the objective is convex, but not strongly convex, Proposition 1 gives a bias term $O(\sigma^2 k/\kappa)$ that increases linearly with $k$. Yet, the following corollary exhibits an optimal rate with finite horizon, when both $\sigma^2$ and an upper-bound on $\|x_0 - x^\star\|^2$ are available. Even though non-practical, the result shows that our analysis recovers the optimal dependency in the noise level, as [18, 28] and others.

**Corollary 3** (Complexity of proximal stochastic gradient algorithm, $\mu = 0$)**.** *Consider a fixed budget $K$ of iterations of Algorithm 1 with $h_k$ defined in (10). When $\kappa = \max(L, \sigma(K+1)^{3/2}/\|x_0 - x^\star\|)$,*

$$\mathbb{E}[F(x_K) - F^\star] \leq \frac{2L\|x_0 - x^\star\|^2}{(K+1)^2} + \frac{3\sigma\|x_0 - x^\star\|}{\sqrt{K+1}}.$$

While all the previous examples use the choice $x_k = x_k^\star$, we will see in Section 4.2 cases where we may choose $x_k \neq x_k^\star$. Before that, we introduce a variant when $x_k^\star$ is not available.

In principle, it is possible to design other surrogates, which would lead to new algorithms coming with convergence guarantees given by Propositions 1 and 4, but the given examples (4), (10), and (10) already cover all important cases considered in the paper for functions of the form (1).

## 4.1  Variant with Inexact Minimization

In this variant, presented in Algorithm 2, $x_k^\star$ is not available and we assume that $\mathcal{M}$ also satisfies:

$(\mathcal{H}_4)$  given $\varepsilon_k \geq 0$, $\mathcal{M}$ can provide a point $x_k$ such that $\mathbb{E}[h_k(x_k) - h_k^\star] \leq \varepsilon_k$.

---

**Algorithm 2** Generic Acceleration Framework with Inexact Minimization of $h_k$

1: **Input:** same as Algorithm 2;
2: **Initialization:** $y_0 = x_0$; $q = \frac{\mu}{\mu+\kappa}$; $\alpha_0 = 1$ if $\mu = 0$ or $\alpha_0 = \sqrt{q}$ if $\mu \neq 0$;
3: **for** $k = 1, \dots, K$ **do**
4:    Consider a surrogate $h_k$ satisfying $(\mathcal{H}_1)$, $(\mathcal{H}_2)$ and obtain $x_k$ satisfying $(\mathcal{H}_4)$;
5:    Compute $\alpha_k$ in $(0,1)$ by solving the equation $\alpha_k^2 = (1 - \alpha_k)\alpha_{k-1}^2 + q\alpha_k$.
6:    Update the extrapolated sequence $y_k = x_k + \beta_k(x_k - x_{k-1})$ with $\beta_k$ defined in (7);
7: **end for**
8: **Output:** $x_k$ (final estimate).

---

The next proposition, proven in Appendix C, gives us some insight on how to achieve acceleration.

**Proposition 4** (Convergence analysis for Algorithm 2)**.** *Consider Alg. 2. Then, for any* $\gamma \in (0,1]$,

$$\mathbb{E}[F(x_k)-F^\star] \leq \begin{cases} \left(1 - \frac{\sqrt{q}}{2}\right)^k \left(2(F(x_0) - F^\star) + 4\sum_{j=1}^{k}\left(1 - \frac{\sqrt{q}}{2}\right)^{-j}\left(\delta_j + \frac{\varepsilon_j}{\sqrt{q}}\right)\right) & \text{if } \mu \neq 0 \\ \frac{2e^{1+\gamma}}{(k+1)^2}\left(\kappa\|x_0 - x^\star\|^2 + \sum_{j=1}^{k}(j+1)^2\delta_j + \frac{(j+1)^{3+\gamma}\varepsilon_j}{\gamma}\right) & \text{if } \mu = 0. \end{cases}$$

To maintain the accelerated rate, the sequence $(\delta_k)_{k \geq 0}$ needs to converge at a similar speed as in Proposition 1, but the dependency in $\varepsilon_k$ is slightly worse. Specifically, when $\mu$ is positive, we may have both $(\varepsilon_k)_{k\geq 0}$ and $(\delta_k)_{k \geq 0}$ decreasing at a rate $O((1 - \rho)^k)$ with $\rho < \sqrt{q}/2$, but we pay a factor $(1/\sqrt{q})$ compared to (8). When $\mu = 0$, the accelerated $O(1/k^2)$ rate is preserved whenever $\varepsilon_k = O(1/k^{4+2\gamma})$ and $\delta_k = O(1/k^{3+\gamma})$, but we pay a factor $O(1/\gamma)$ compared to (8).

**Catalyst [33].**  When using $h_k$ defined in (4), we recover the convergence rates of [33]. In such a case $\delta_k = \varepsilon_k$ since $\mathbb{E}[F(x_k)] \leq \mathbb{E}[h_k(x_k)] \leq \mathbb{E}[h_k^\star] + \delta_k$. In order to analyze the complexity of minimizing each $h_k$ with $\mathcal{M}$ and derive the global complexity of the multi-stage algorithm, the next proposition, proven in Appendix C, characterizes the quality of the initialization $x_{k-1}$.

**Proposition 5** (Warm restart for Catalyst)**.** *Consider Alg. 2 with $h_k$ defined in (4). Then, for $k \geq 2$,*

$$\mathbb{E}[h_k(x_{k-1}) - h_k^\star] \leq \frac{3\varepsilon_{k-1}}{2} + 54\kappa \max\left(\|x_{k-1} - x^\star\|^2, \|x_{k-2} - x^\star\|^2, \|x_{k-3} - x^\star\|^2\right), \qquad (12)$$

where $x_{-1} = x_0$. Following [33], we may now analyze the global complexity. For instance, when $f$ is $\mu$-strongly convex, we may choose $\varepsilon_k = O((1 - \rho)^k(F(x_0) - F^\star))$ with $\rho = \sqrt{q}/3$. Then, it is possible to show that Proposition (4) yields $\mathbb{E}[F(x_k) - F^\star] = O(\varepsilon_k/q)$ and from the inequality $\frac{\mu}{2}\|x_k - x^\star\|^2 \leq F(x_k) - F^\star$ and (12), we have $\mathbb{E}[h_k(x_{k-1}) - h_k^\star] = O(\frac{\kappa}{\mu q}\varepsilon_{k-1}) = O(\varepsilon_{k-1}/q^2)$. Consider now a method $\mathcal{M}$ that behaves as (2). When $\sigma = 0$, $x_k$ can be obtained in $O(\log(1/q)/\tau) = \tilde{O}(1/\tau)$ iterations of $\mathcal{M}$ after initializing with $x_{k-1}$. This allows us to obtain the global complexity $\tilde{O}((1/\tau\sqrt{q})\log(1/\varepsilon))$. For example, when $\mathcal{M}$ is the proximal gradient descent method, $\kappa = L$ and $\tau = (\mu + \kappa)/(L + \kappa)$ yield the global complexity $\tilde{O}(\sqrt{L/\mu}\log(1/\varepsilon))$ of an accelerated method.

Our results improve upon Catalyst [33] in two aspects that are crucial for stochastic optimization: (i) we allow the sub-problems to be solved in expectation, whereas Catalyst requires the stronger condition $h_k(x_k) - h_k^\star \leq \varepsilon_k$; (ii) Proposition 5 removes the requirement of [33] to perform a full gradient step for initializing the method $\mathcal{M}$ in the composite case (see Prop. 12 in [33]).

**Proximal gradient descent with inexact prox [45].**  The surrogate (10) with inexact minimization can be treated in the same way as Catalyst, which provides a unified proof for both problems. Then, we recover the results of [45], while allowing inexact minimization to be performed in expectation.

**Stochastic Catalyst.**  With Proposition 5, we are in shape to consider stochastic problems when using a method $\mathcal{M}$ that converges linearly as (2) with $\sigma^2 \neq 0$ for minimizing $h_k$. As in Section 3,

we also assume that there exists a mini-batch/step-size parameter $\eta$ that can reduce the bias by a factor $\eta < 1$ while paying a factor $1/\eta$ in terms of inner-loop complexity. As above, we discuss the strongly-convex case and choose the same sequence $(\varepsilon_k)_{k \geq 0}$. In order to minimize $h_k$ up to accuracy $\varepsilon_k$, we set $\eta_k = \min(1, \varepsilon_k/(2B\sigma^2))$ such that $\eta_k B\sigma^2 \leq \varepsilon_k/2$. Then, the complexity to minimize $h_k$ with $\mathcal{M}$ when using the initialization $x_{k-1}$ becomes $\tilde{O}(1/\tau\eta_k)$, leading to the global complexity

$$\tilde{O}\left(\frac{1}{\tau\sqrt{q}}\log\left(\frac{F(x_0) - F^\star}{\varepsilon}\right) + \frac{B\sigma^2}{q^{3/2}\tau\varepsilon}\right). \tag{13}$$

Details about the derivation are given in Appendix B. The left term corresponds to the Catalyst accelerated rate, but it may be shown that the term on the right is sub-optimal. Indeed, consider $\mathcal{M}$ to be ISTA with $\kappa = L - \mu$. Then, $B = 1/L$, $\tau = O(1)$, and the right term becomes $\tilde{O}((\sqrt{L/\mu})\sigma^2/\mu\varepsilon)$, which is sub-optimal by a factor $\sqrt{L/\mu}$. Whereas this result is a negative one, suggesting that Catalyst is not robust to noise, we show in Section 4.2 how to circumvent this for a large class of algorithms.

**Accelerated stochastic proximal gradient descent with inexact prox.** Finally, consider $h_k$ defined in (10) but the proximal operator is computed approximately, which, to our knowledge, has never been analyzed in the stochastic context. Then, it may be shown (see Appendix B for details) that, even though $x_k^\star$ is not available, Proposition 4 holds nonetheless with $\delta_k = 2\varepsilon_k + 3\sigma^2/(2(\kappa + \mu))$. Then, an interesting question is how small should $\varepsilon_k$ be to guarantee the optimal dependency with respect to $\sigma^2$ as in Corollary 2. In the strongly-convex case, Proposition 4 simply gives $\varepsilon_k = O(\sqrt{q}\sigma^2/(\kappa + \mu))$ such that $\delta_k \approx \varepsilon_k/\sqrt{q}$.

## 4.2 Exploiting methods $\mathcal{M}$ providing strongly convex surrogates

Among various application cases, we have seen an extension of Catalyst to stochastic problems. To achieve convergence, the strategy requires a mechanism to reduce the bias $B\sigma^2$ in (2), *e.g.*, by using mini-batches or decreasing step sizes. Yet, the approach suffers from two issues: (i) some of the parameters are based on unknown quantities such as $\sigma^2$; (ii) the worst-case complexity exhibits a sub-optimal dependency in $\sigma^2$, typically of order $1/\sqrt{q}$ when $\mu > 0$. Whereas practical workarounds for the first point are discussed in Section 5, we now show how to solve the second one in some cases, by using Algorithm 1 with an optimization method $\mathcal{M}$, which is able not only to minimize an auxiliary objective $H_k$, but also at the same time is able to provide a model $h_k$, typically a quadratic function, which is easy to minimize. Consider then a method $\mathcal{M}$ satisfying (2) and which produces, after $T$ steps, a point $x_k$ and a surrogate $h_k$ such that

$$\mathbb{E}[H_k(x_k) - h_k^\star] \leq C(1-\tau)^T(H_k(x_{k-1}) - H_k^\star + \xi_{k-1}) + B\sigma^2 \quad \text{with} \quad H_k(x) = F(x) + \frac{\kappa}{2}\|x - y_{k-1}\|^2,$$
$$\tag{14}$$

where $H_k$ is approximately minimized by $\mathcal{M}$, $h_k$ is a model of $H_k$ that satisfies $(\mathcal{H}_1)$, $(\mathcal{H}_2)$ and that can be minimized in closed form, and $\xi_{k-1} = O(\mathbb{E}[F(x_{k-1}) - F^\star])$; it is easy to show that $(\mathcal{H}_3)$ is also satisfied with the choice $\delta_k = C(1 - \tau)^T(H_k(x_{k-1}) - H_k^\star + \xi_{k-1}) + B\sigma^2$ since $\mathbb{E}[F(x_k)] \leq \mathbb{E}[H_k(x_k)] \leq \mathbb{E}[h_k^\star] + \delta_k$. In other words, $\mathcal{M}$ is used to perform *approximate minimization* of $H_k$, but we consider cases where $\mathcal{M}$ also provides *another surrogate* $h_k$ with closed-form minimizer that satisfies the conditions required to use Algorithm 1, which has better convergence guarantees than Algorithm 2 (same convergence rate up to a better factor).

As shown in Appendix D, even though (14) looks technical, a large class of optimization techniques are able to provide the condition (14), including many variants of proximal stochastic gradient descent methods with variance reduction such as SAGA [13], MISO [35], SDCA [47], or SVRG [53].

Whereas (14) seems to be a minor modification of (2), an important consequence is that it will allow us to gain a factor $1/\sqrt{q}$ in complexity when $\mu > 0$, corresponding precisely to the sub-optimality factor. Therefore, even though the surrogate $H_k$ needs only be minimized approximately, the condition (14) allows us to use Algorithm 1 instead of Algorithm 2. The dependency with respect to $\delta_k$ being better than $\varepsilon_k$ (by $1/\sqrt{q}$), we have then the following result:

**Proposition 6** (Stochastic Catalyst with Optimality Gaps, $\mu > 0$). *Consider Algorithm 1 with a method $\mathcal{M}$ and surrogate $h_k$ satisfying (14) when $\mathcal{M}$ is used to minimize $H_k$ by using $x_{k-1}$ as a warm restart. Assume that $f$ is $\mu$-strongly convex and that there exists a parameter $\eta$ that can reduce the bias $B\sigma^2$ by a factor $\eta < 1$ while paying a factor $1/\eta$ in terms of inner-loop complexity.*

*Choose $\delta_k = O((1-\sqrt{q}/2)^k(F(x_0)-F^\star))$ and $\eta_k = \min(1, \delta_k/(2B\sigma^2))$. Then, the complexity to solve (14) and compute $x_k$ is $\tilde{O}(1/\tau\eta_k)$, and the global complexity to obtain $\mathbb{E}[F(x_k)-F^\star] \leq \varepsilon$ is*

$$\tilde{O}\left(\frac{1}{\tau\sqrt{q}}\log\left(\frac{F(x_0)-F^\star}{\varepsilon}\right) + \frac{B\sigma^2}{q\tau\varepsilon}\right).$$

The term on the left is the accelerated rate of Catalyst for deterministic problems, whereas the term on the right is potentially optimal for strongly convex problems, as illustrated in the next table. We provide indeed practical choices for the parameters $\kappa$, leading to various values of $B, \tau, q$, for the proximal stochastic gradient descent method with iterate averaging as well as variants of SAGA,MISO,SVRG that can cope with stochastic perturbations, which are discussed in Appendix D. All the values below are given up to universal constants to simplify the presentation.

| Method $\mathcal{M}$ | $h_k$ | $\kappa$ | $\tau$ | $B$ | $q$ | Complexity after Catalyst |
|---|---|---|---|---|---|---|
| prox-SGD | (10) | $L-\mu$ | $\frac{1}{2}$ | $\frac{1}{L}$ | $\frac{\mu}{L}$ | $\tilde{O}\left(\sqrt{\frac{L}{\mu}}\log\left(\frac{F_0}{\varepsilon}\right) + \frac{\sigma^2}{\mu\varepsilon}\right)$ |
| SAGA/MISO/SVRG with $\frac{L}{n} \geq \mu$ | (14) | $\frac{L}{n}-\mu$ | $\frac{1}{n}$ | $\frac{1}{L}$ | $\frac{\mu n}{L}$ | $\tilde{O}\left(\sqrt{n\frac{L}{\mu}}\log\left(\frac{F_0}{\varepsilon}\right) + \frac{\sigma^2}{\mu\varepsilon}\right)$ |

In this table, $F_0 := F(x_0)-F^\star$ and the methods SAGA/MISO/SVRG are applied to the stochastic finite-sum problem discussed in Section 1 with $n$ $L$-smooth functions. As in the deterministic case, we note that when $L/n \leq \mu$, there is no acceleration for SAGA/MISO/SVRG since the complexity of the unaccelerated method $\mathcal{M}$ is $\tilde{O}\left(n\log\left(F_0/\varepsilon\right) + \sigma^2/\mu\varepsilon\right)$, which is independent of the condition number and already optimal [28]. In comparison, the logarithmic terms in $L, \mu$ that are hidden in the notation $\tilde{O}$ do not appear for a variant of the SVRG method with direct acceleration introduced in [28]. Here, our approach is more generic. Note also that $\sigma^2$ for prox-SGD and SAGA/MISO/SVRG cannot be compared to each other since the source of randomness is larger for prox-SGD, see [7, 28].

## 5 Experiments

In this section, we perform numerical evaluations by following [28], which was notably able to make SVRG and SAGA robust to stochastic noise, and accelerate SVRG. Code to reproduce the experiments is provided with the submission and more details and experiments are given in Appendix E.

**Formulations.** Given training data $(a_i, b_i)_{i=1,\ldots,n}$, with $a_i$ in $\mathbb{R}^p$ and $b_i$ in $\{-1, +1\}$, we consider the optimization problem

$$\min_{x \in \mathbb{R}^p} \frac{1}{n}\sum_{i=1}^{n} \phi(b_i a_i^\top x) + \frac{\mu}{2}\|x\|^2,$$

where $\phi$ is either the logistic loss $\phi(u) = \log(1+e^{-u})$, or the squared hinge loss $\phi(u) = \frac{1}{2}\max(0, 1-u)^2$, which are both $L$-smooth, with $L = 0.25$ for logistic and $L = 1$ for the squared hinge loss. Studying the squared hinge loss is interesting since its gradients are unbounded on the optimization domain, which may break the bounded noise assumption. The regularization parameter $\mu$ acts as the strong convexity constant for the problem and is chosen among the smallest values one would try when performing parameter search, *e.g.*, by cross validation. Specifically, we consider $\mu = 1/10n$ and $\mu = 1/100n$, where $n$ is the number of training points; we also try $\mu = 1/1000n$ to evaluate the numerical stability of methods in very ill-conditioned problems. Following [7, 28, 54], we consider DropOut perturbations [51]—that is, setting each component $(\nabla f(x))_i$ to 0 with a probability $\delta$ and to $(\nabla f(x))_i/(1 - \delta)$ otherwise. This procedure is motivated by the need of a simple optimization benchmark illustrating stochastic finite-sum problems, where the amount of perturbation is easy to control. The settings used in our experiments are $\delta = 0$ (no noise) and $\delta \in \{0.01, 0.1\}$.

**Datasets.** We consider three datasets with various number of points $n$ and dimension $p$. All the data points are normalized to have unit $\ell_2$-norm. The description comes from [28]:

- alpha is from the Pascal Large Scale Learning Challenge website[2] and contains $n = 250\,000$ points in dimension $p = 500$.

- gene consists of gene expression data and the binary labels $b_i$ characterize two different types of breast cancer. This is a small dataset with $n = 295$ and $p = 8\,141$.
- ckn-cifar is an image classification task where each image from the CIFAR-10 dataset[3] is represented by using a two-layer unsupervised convolutional neural network [36]. We consider here the binary classification task consisting of predicting the class 1 vs. other classes, and use our algorithms for the classification layer of the network, which is convex. The dataset contains $n = 50\,000$ images and the dimension of the representation is $p = 9\,216$.

**Methods.** We consider the variants of SVRG and SAGA of [28], which use decreasing step sizes when $\delta > 0$ (otherwise, they do not converge). We use the suffix "-d" each time decreasing step sizes are used. We also consider Katyuasha [1] when $\delta = 0$, and the accelerated SVRG method of [28], denoted by acc-SVRG. Then, SVRG-d, SAGA-d, acc-SVRG-d are used with the step size strategies described in [28], by using the code provided to us by the authors.

**Practical questions and implementation.** In all setups, we choose the parameter $\kappa$ according to theory, which are described in the previous section, following Catalyst [33]. For composite problems, Proposition 5 suggests to use $x_{k-1}$ as a warm start for inner-loop problems. For smooth ones, [33] shows that in fact, other choices such as $y_{k-1}$ are appropriate and lead to similar complexity results. In our experiments with smooth losses, we use $y_{k-1}$, which has shown to perform consistently better.

The strategy for $\eta_k$ discussed in Proposition 6 suggests to use constant step-sizes for a while in the inner-loop, typically of order $1/(\kappa + L)$ for the methods we consider, before using an exponentially decreasing schedule. Unfortunately, even though theory suggests a rate of decay in $(1 - \sqrt{q}/2)^k$, it does not provide useful insight on when decaying should start since the theoretical time requires knowing $\sigma^2$. A similar issue arise in stochastic optimization techniques involving iterate averaging [9]. We adopt a similar heuristic as in this literature and start decaying after $k_0$ epochs, with $k_0 = 30$. Finally, we discuss the number of iterations of $\mathcal{M}$ to perform in the inner-loop. When $\eta_k = 1$, the theoretical value is of order $\tilde{O}(1/\tau) = \tilde{O}(n)$, and we choose exactly $n$ iterations (one epoch), as in Catalyst [33]. After starting decaying the step-sizes ($\eta_k < 1$), we use $\lceil n/\eta_k \rceil$, according to theory.

**Experiments and conclusions.** We run each experiment five time with a different random seed and average the results. All curves also display one standard deviation. Appendix E contains numerous experiments, where we vary the amount of noise, the type of approach (SVRG vs. SAGA), the amount of regularization $\mu$, and choice of loss function. In Figure 1, we show a subset of these curves. Most of them show that acceleration may be useful even in the stochastic optimization regime, consistently with [28]. At the same time, all acceleration methods may not perform well for very ill-conditioned problems with $\mu = 1/1000n$, where the sublinear convergence rates for convex optimization ($\mu = 0$) are typically better than the linear rates for strongly convex optimization ($\mu > 0$). However, these ill-conditioned cases are often unrealistic in the context of empirical risk minimization.

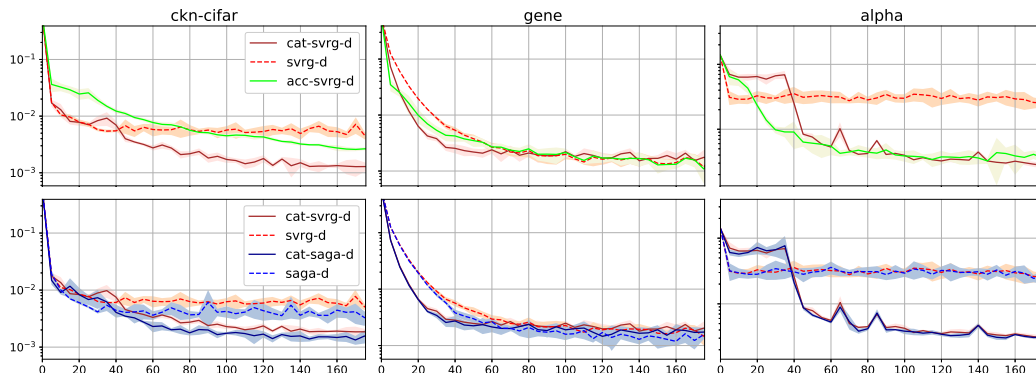

Figure 1: Accelerating SVRG-like (top) and SAGA (bottom) methods for $\ell_2$-logistic regression with $\mu = 1/(100n)$ (bottom) for $\delta = 0.1$. All plots are on a logarithmic scale for the objective function value, and the $x$-axis denotes the number of epochs. The colored tubes around each curve denote a standard deviations across 5 runs. They do not look symmetric because of the logarithmic scale.

**Acknowledgments**

This work was supported by the ERC grant SOLARIS (number 714381) and ANR 3IA MIAI@Grenoble Alpes. The authors would like to thank Anatoli Juditsky for numerous interesting discussions that greatly improved the quality of this manuscript.

## Footnotes

[1] All objectives addressed by the original Catalyst approach are deterministic, even though they may be large finite sums. Here, we consider general expectations as defined in (1).

[2] `http://largescale.ml.tu-berlin.de/`

[3] https://www.cs.toronto.edu/~kriz/cifar.html

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
