[Supplementary Material · supplementary.pdf]

# A  Useful Results and Definitions

In this section, we present auxiliary results and definitions.

**Definition 7** (Proximal operator). *Given a convex lower-semicontinuous function $\psi$ defined on $\mathbb{R}^p$, the proximal operator of $\psi$ is defined as the unique solution of the strongly-convex problem*

$$Prox_\psi[y] = \operatorname*{argmin}_{x \in \mathbb{R}^p} \left\{ \frac{1}{2} \|y - x\|^2 + \psi(x) \right\}.$$

**Lemma 8** (Convergence rate of the sequences $(\alpha_k)_{k \geq 0}$ and $(A_k)_{k \geq 0}$). *Consider the sequence in $(0, 1)$ defined by the recursion*

$$\alpha_k^2 = (1 - \alpha_k)\alpha_{k-1}^2 + q\alpha_k \quad with \quad 0 \leq q < 1,$$

*and define $A_k = \prod_{t=1}^{k}(1 - \alpha_t)$. Then,*

- *if $q = 0$ and $\alpha_0 = 1$, then, for all $k \geq 1$,*

$$\frac{2}{(k+2)^2} \leq A_k = \alpha_k^2 \leq \frac{4}{(k+2)^2}.$$

- *if $\alpha_0 = \sqrt{q}$, then for all $k \geq 1$,*

$$A_k = (1 - \sqrt{q})^k \quad and \quad \alpha_k = \sqrt{q}.$$

- *if $\alpha_0 = 1$, then for all $k \geq 1$,*

$$A_k \leq \min\left((1 - \sqrt{q})^k, \frac{4}{(k+2)^2}\right) \quad and \quad \alpha_k \geq \max\left(\sqrt{q}, \frac{\sqrt{2}}{k+2}\right).$$

*Proof.* We prove the three points, one by one.

**First point.** Let us prove the first point when $q = 0$ and $\alpha_0 = 1$. The relation $A_k = \alpha_k^2$ is obvious for all $k \geq 1$ and the relation $\alpha_k^2 \leq \frac{4}{(k+2)^2}$ holds for $k = 0$. By induction, let us assume that we have the relation $\alpha_{k-1}^2 \leq \frac{4}{(k+1)^2}$ and let us show that it propagates for $\alpha_k^2$. Assume, by contradiction, that $\alpha_k^2 > \frac{4}{(k+2)^2}$, meaning that $\alpha_k > \frac{2}{(k+2)}$. Then,

$$\alpha_k^2 = (1 - \alpha_k)\alpha_{k-1}^2 \leq (1 - \alpha_k)\frac{4}{(k+1)^2} < \frac{4k}{(k+2)(k+1)^2} = \frac{4}{(k+2)(k+2+\frac{1}{k})} < \frac{4}{(k+2)^2},$$

and we obtain a contradiction. Therefore, $\alpha_k^2 \leq \frac{4}{(k+2)^2}$ and the induction hypothesis allows us to conclude for all $k \geq 1$. Then, note [44] that we also have for all $k \geq 1$,

$$A_k = \prod_{t=1}^{k}(1 - \alpha_t) \geq \prod_{t=1}^{k}\left(1 - \frac{2}{t+2}\right) = \frac{2}{(k+1)(k+2)} \geq \frac{2}{(k+2)^2}.$$

**Second point.** The second point is obvious by induction.

**Third point.** For the third point, we simply assume $\alpha_0 = 1$ such that $\alpha_0 \geq \sqrt{q}$. Then, the relation $\alpha_k \geq \sqrt{q}$ and therefore $A_k \leq (1 - \sqrt{q})^k$ are easy to show by induction. Then, consider the sequence defined recursively by $u_k^2 = (1 - u_k)u_{k-1}^2$ with $u_0 = 1$. From the first point, we have that $\frac{\sqrt{2}}{k+2} \leq u_k \leq \frac{2}{k+2}$. We will show that $\alpha_k \geq u_k$ for all $k \geq 0$, which will be sufficient to conclude since then we would have $A_k \leq \prod_{t=1}^{k}(1 - u_t) \leq \frac{4}{(k+2)^2}$. First, we note that $\alpha_0 = u_0$; then, assume that $\alpha_{k-1} \geq u_{k-1}$ and also assume by contradiction that $\alpha_k > u_k$. This implies that

$$u_k^2 = (1 - u_k)u_{k-1}^2 \leq (1 - u_k)\alpha_{k-1}^2 < (1 - \alpha_k)\alpha_{k-1}^2 \leq \alpha_k^2,$$

which contradicts the assumption $\alpha_k > u_k$. This allows us to conclude by induction.  $\square$

**Lemma 9** (Convergence rate of sequences $\Theta_k = \prod_{i=1}^{k}(1 - \theta_i)$). *Consider the sequence* $\theta_j = \frac{\gamma}{(1+j)^{1+\gamma}}$ *with* $\gamma$ *in* $(0, 1]$. *Then,*

$$e^{-(1+\gamma)} \leq \Theta_k \leq 1. \tag{15}$$

*Proof.* We use the classical inequality $\log(1 + u) \geq \frac{u}{1+u}$ for all $u > -1$:

$$-\log(\Theta_k) = -\sum_{j=1}^{k} \log\left(1 - \frac{\gamma}{(1+j)^{1+\gamma}}\right) \leq \sum_{j=1}^{k} \frac{\gamma}{(1+j)^{1+\gamma} - \gamma} \leq \sum_{j=1}^{k} \frac{\gamma}{j^{1+\gamma}},$$

when noting that the function $g(x) = (1+x)^{1+\gamma} - x^{1+\gamma}$ is greater than $\gamma$ for all $x \geq 1$, since $g(1) \geq 1 \geq \gamma$ and $g$ is non-decreasing. Then,

$$-\log(\Theta_k) \leq \sum_{j=1}^{k} \frac{\gamma}{j^{1+\gamma}} \leq \gamma + \gamma \int_{x=1}^{k} \frac{1}{x^{1+\gamma}} dx = \gamma + 1 - \frac{1}{k^\gamma} \leq \gamma + 1.$$

Then, we immediately obtain (15).

$\square$

# B    Details about Complexity Results

## B.1    Details about (5)

Consider the complexity (2) with $h = F$. To achieve the accuracy $2B\sigma^2$, it is sufficient to run the method $\mathcal{M}$ for $t_0$ iterations, such that

$$C(1 - \tau)^{t_0}(F(x_0) - F^\star) \leq B\sigma^2.$$

It is then easy to see that this inequality is satisfied as soon as $t_0$ is greater than $\frac{1}{\tau}\log(C(F(x_0) - F^\star)/B\sigma^2)$. Since $\varepsilon \leq B\sigma^2$ and using the concavity of the logarithm function, it is also sufficient to choose $t_0 = \frac{1}{\tau}\log(C(F(x_0) - F^\star)/\varepsilon)$.

Then, we perform $K$ restart stages such that $\varepsilon_K \leq \varepsilon$. Each stage is initialized with a point $x_k$ satisfying $\mathbb{E}[F(x_k) - F^\star] \leq \varepsilon_{k-1}$, and the goal of each stage is to reduce the error by a factor $1/2$. Given that $\eta_k$ increases the computational cost, the complexity of the $k$-th stage is then upper-bounded by $\frac{2^k}{\tau}\log(2C)$, leading to the global complexity

$$O\left(\frac{1}{\tau}\log\left(\frac{C(F(x_0) - F^\star)}{\varepsilon}\right) + \sum_{k=1}^{K} \frac{2^k}{\tau}\log(2C)\right) \quad \text{with} \quad K = \left\lceil \log_2\left(\frac{2B\sigma^2}{\varepsilon}\right)\right\rceil,$$

and (5) follows by elementary calculations.

## B.2    Obtaining (5) from (6)

Since $h$ is $\mu$-strongly convex, we notice that (6) implies the rate

$$\mathbb{E}[h(z_t) - h^\star] \leq \frac{D(h(z_0) - h^\star)}{\mu t^d} + \frac{B\sigma^2}{2},$$

by using the strong convexity inequality $h(z_0) \geq h^\star + \frac{\mu}{2}\|z_0 - z^\star\|^2$. After running the algorithm for $t' = \lceil (2D/\mu)^{1/d}\rceil$ iterations, we can show that

$$\mathbb{E}[h(z_{t'}) - h^\star] \leq \frac{h(z_0) - h^\star}{2} + \frac{B\sigma^2}{2}.$$

Then, when restarting the procedure $s$ times (using the solution of the previous iteration as initialization), and denoting by $h_{st'}$ the last iterate, it is easy to show that

$$\mathbb{E}[h(x_{st'}) - h^\star] \leq \frac{h(x_0) - h^\star}{2^s} + \frac{B\sigma^2}{2}\left(\sum_{i=0}^{s-1} \frac{1}{2^i}\right) \leq \frac{h(z_0) - h^\star}{2^s} + B\sigma^2.$$

Then, calling $t = st'$, we can use the inequality $2^{-u} \leq 1 - \frac{u}{2}$ for $u$ in $[0, 1]$, due to convexity, and

$$\mathbb{E}[h(z_t) - h^\star] \leq (h(z_0) - h^\star)\left(2^{-1/t'}\right)^t + B\sigma^2 = (h(z_0) - h^\star)\left(1 - \frac{1}{2t'}\right)^k + B\sigma^2,$$

which gives us (2) with $C = 1$ and $\tau = \frac{1}{2t'}$. It is then easy to obtain (5) by following similar steps as in Section B.1, by noticing that the restart frequency is of the same order $O(1/\tau)$.

## B.3 Details about (13)

**Inner-loop complexity.** Since $\eta_k$ is chosen such that the bias $\eta_k B\sigma^2$ is smaller than $\varepsilon_k$, the number of iterations of $\mathcal{M}$ to solve the sub-problem is $\tilde{O}(\tau) = O(\log(1/q)\tau)$, as in the deterministic case, and the complexity is thus $\tilde{O}(\tau/\eta_k)$.

**Outer-loop complexity.** Since $\mathbb{E}[F(x_k) - F^\star] \leq O((1 - \sqrt{q}/3)^k(F(x_0) - F^\star))/q$ according to Proposition 4, it suffices to choose

$$K = O\left(\frac{1}{\sqrt{q}}\log\left(\frac{F(x_0) - F^\star}{q\varepsilon}\right)\right)$$

iterations to guarantee $\mathbb{E}[F(x_K) - F^\star] \leq \varepsilon = O(\varepsilon_K/q) = O((1 - \sqrt{q}/3)^K(F(x_0) - F^\star)/q)$.

**Global complexity.** The total complexity to guarantee $\mathbb{E}[F(x_k) - F^\star] \leq \varepsilon$ is then

$$
\begin{aligned}
C &= \sum_{k=1}^{K} \tilde{O}\left(\frac{\tau}{\eta_k}\right) \\
&\leq \tilde{O}\left(\sum_{k=1}^{K}\tau + \sum_{k=1}^{K}\frac{B\sigma^2\tau}{\varepsilon_k}\right) \\
&= \tilde{O}\left(\sum_{k=1}^{K}\tau + \sum_{k=1}^{K}\frac{B\sigma^2\tau}{\left(1 - \frac{\sqrt{q}}{3}\right)^k(F(x_0) - F^\star)}\right) \\
&= \tilde{O}\left(\frac{\tau}{\sqrt{q}}\log\left(\frac{F(x_0) - F^\star}{\varepsilon}\right) + \frac{B\sigma^2\tau}{\sqrt{q}\left(1 - \frac{\sqrt{q}}{3}\right)^{K+1}(F(x_0) - F^\star)}\right) \\
&= \tilde{O}\left(\frac{\tau}{\sqrt{q}}\log\left(\frac{F(x_0) - F^\star}{\varepsilon}\right) + \frac{B\sigma^2\tau}{q^{3/2}\varepsilon}\right),
\end{aligned}
$$

where the last relation uses the fact that $\varepsilon = O(\varepsilon_K/q) = O((1 - \sqrt{q}/3)^K(F(x_0) - F^\star)/q)$.

## B.4 Complexity of accelerated stochastic proximal gradient descent with inexact prox

Assume that $h_k(x_k) - h_k^\star \leq \varepsilon_k$. Then, following similar steps as in (11),

$$
\begin{aligned}
\mathbb{E}[F(x_k)] &\leq \mathbb{E}[h_k(x_k)] + \mathbb{E}[(g_k - \nabla f(y_{k-1}))^\top(x_k - y_{k-1})] \\
&= \mathbb{E}[h_k(x_k)] + \mathbb{E}[(g_k - \nabla f(y_{k-1}))^\top(x_k - u_{k-1})] \\
&= \mathbb{E}[h_k(x_k)] + \mathbb{E}[(g_k - \nabla f(y_{k-1}))^\top(x_k - x_k^\star)] + \mathbb{E}[(g_k - \nabla f(y_{k-1}))^\top(x_k^\star - u_{k-1})] \\
&\leq \mathbb{E}[h_k(x_k)] + \mathbb{E}[(g_k - \nabla f(y_{k-1}))^\top(x_k - x_k^\star)] + \frac{\sigma^2}{\kappa + \mu} \\
&\leq \mathbb{E}[h_k(x_k)] + \frac{\mathbb{E}[\|g_k - \nabla f(y_{k-1})\|^2]}{2(\kappa + \mu)} + \frac{(\kappa + \mu)\mathbb{E}[\|x_k - x_k^\star\|^2]}{2} + \frac{\sigma^2}{\kappa + \mu} \\
&\leq \mathbb{E}[h_k(x_k)] + \mathbb{E}[h_k(x_k) - h_k^\star] + \frac{3\sigma^2}{2(\kappa + \mu)} \\
&\leq \mathbb{E}[h_k^\star] + 2\varepsilon_k + \frac{3\sigma^2}{2(\kappa + \mu)}.
\end{aligned}
$$

And thus, $\delta_k = 2\varepsilon_k + \frac{3\sigma^2}{2(\kappa+\mu)}$.

## C  Proofs of Main Results

### C.1  Proof of Propositions 1 and 4

*Proof.* In order to treat both propositions jointly, we introduce the quantity

$$w_k = \begin{cases} x_k^\star & \text{for Algorithm 1} \\ x_k & \text{for Algorithm 2} \end{cases},$$

and, for all $k \geq 1$,

$$v_k = w_k + \frac{1 - \alpha_{k-1}}{\alpha_{k-1}}(w_k - x_{k-1}), \tag{16}$$

with $v_0 = x_0$, as well as $\eta_k = \frac{\alpha_k - q}{1 - q}$ for all $k \geq 0$.

Note that the following relations hold for all $k \geq 1$, keeping in mind that $\alpha_k^2 = (1 - \alpha_k)\alpha_{k-1}^2 + q\alpha_k$:

$$1 - \eta_k = \frac{1 - \alpha_k}{1 - q} = \frac{(\kappa + \mu)(1 - \alpha_k)}{\kappa}$$

$$\eta_k = \frac{\alpha_k - q}{1 - q} = \frac{\alpha_k^2 - q\alpha_k}{\alpha_k - q\alpha_k} = \frac{\alpha_{k-1}^2(1 - \alpha_k)}{\alpha_k - \alpha_k^2 + (1 - \alpha_k)\alpha_{k-1}^2} = \frac{\alpha_{k-1}^2}{\alpha_{k-1}^2 + \alpha_k}.$$

Then, based on the previous relations, we have

$$\begin{aligned} y_k &= w_k + \beta_k(w_k - x_{k-1}) + \frac{(\kappa + \mu)(1 - \alpha_k)}{\kappa}(x_k - w_k) \\ &= w_k + \frac{\alpha_{k-1}(1 - \alpha_{k-1})}{\alpha_{k-1}^2 + \alpha_k}(w_k - x_{k-1}) + (1 - \eta_k)(x_k - w_k) \\ &= w_k + \frac{\eta_k(1 - \alpha_{k-1})}{\alpha_{k-1}}(w_k - x_{k-1}) + (1 - \eta_k)(x_k - w_k) \\ &= \eta_k v_k + (1 - \eta_k)x_k, \end{aligned}$$

which is similar to the relation used in [33] when $w_k = x_k$. Then, the proof differs from [33] since we introduce the surrogate function $h_k$. For all $x$ in $\mathbb{R}^p$,

$$\begin{aligned} h_k(x) &\geq h_k^\star + \frac{\kappa + \mu}{2}\|x - x_k^\star\|^2 \quad \text{(by strong convexity, see } \mathcal{H}_1) \\ &= h_k^\star + \frac{\kappa + \mu}{2}\|x - w_k\|^2 + \underbrace{\frac{\kappa + \mu}{2}\|w_k - x_k^\star\|^2 + (\kappa + \mu)\langle x - w_k, w_k - x_k^\star\rangle}_{-\Delta_k(x)}. \end{aligned} \tag{17}$$

Introduce now the following quantity for the convergence analysis:

$$z_{k-1} = \alpha_{k-1}x^\star + (1 - \alpha_{k-1})x_{k-1},$$

and consider $x = z_{k-1}$ in (17) while taking expectations, noting that all random variables indexed by $k{-}1$ are deterministic given $\mathcal{F}_{k-1}$,

$$\begin{aligned} \mathbb{E}[F(x_k)] &\leq \mathbb{E}[h_k^\star] + \delta_k \quad \text{(by } \mathcal{H}_3) \\ &\leq \mathbb{E}[h_k(z_{k-1})] - \mathbb{E}\left[\frac{\kappa + \mu}{2}\|z_{k-1} - w_k\|^2\right] + \mathbb{E}[\Delta_k(z_{k-1})] + \delta_k \\ &\leq \mathbb{E}[F(z_{k-1})] + \mathbb{E}\left[\frac{\kappa}{2}\|z_{k-1} - y_{k-1}\|^2\right] - \mathbb{E}\left[\frac{\kappa + \mu}{2}\|z_{k-1} - w_k\|^2\right] + \mathbb{E}[\Delta_k(z_{k-1})] + \delta_k, \end{aligned} \tag{18}$$

where the last inequality is due to $(\mathcal{H}_2)$.

Let us now open a parenthesis and derive a few relations that will be useful to find a Lyapunov function. To use more compact notation, define $X_k = \mathbb{E}[\|x^\star - x_k\|^2]$, $V_k = \mathbb{E}[\|x^\star - v_k\|^2]$ and $F_k = \mathbb{E}[F(x_k) - F^\star]$, and note that

$$
\begin{aligned}
\mathbb{E}[F(z_{k-1})] &\leq \alpha_{k-1} f^\star + (1 - \alpha_{k-1}) \mathbb{E}[F(x_{k-1})] - \frac{\mu \alpha_{k-1}(1 - \alpha_{k-1})}{2} X_{k-1} \\
\mathbb{E}[\|z_{k-1} - w_k\|^2] &= \alpha_{k-1}^2 V_k \\
\mathbb{E}[\|z_{k-1} - y_{k-1}\|^2] &\leq \alpha_{k-1}(\alpha_{k-1} - \eta_{k-1}) X_{k-1} + \alpha_{k-1} \eta_{k-1} V_{k-1}.
\end{aligned}
\tag{19}
$$

The first relation is due to the convexity of $f$; the second one can be obtained from the definition of $v_k$ in (16) after simple calculations; the last one can be obtained as in the proof of Theorem 3 in [33] (end of page 16).

We may now come back to (18) and we use the relations (19):

$$
\begin{aligned}
F_k + \frac{(\kappa + \mu)\alpha_{k-1}^2}{2} V_k \leq{} & (1 - \alpha_{k-1}) F_{k-1} - \frac{\mu \alpha_{k-1}(1 - \alpha_{k-1})}{2} X_{k-1} + \\
& \frac{\kappa}{2} \alpha_{k-1}(\alpha_{k-1} - \eta_{k-1}) X_{k-1} + \frac{\kappa}{2} \alpha_{k-1} \eta_{k-1} V_{k-1} + \delta_k + \mathbb{E}[\Delta_k(z_{k-1})].
\end{aligned}
$$

It is then easy to see that the terms involving $X_{k-1}$ cancel each other since $\eta_{k-1} = \alpha_{k-1} - \frac{\mu}{\kappa}(1 - \alpha_{k-1})$.

**Lyapunov function.** We may finally define the Lyapunov function

$$
S_k = (1 - \alpha_k) F_k + \frac{\kappa \alpha_k \eta_k}{2} V_k.
\tag{20}
$$

and we obtain

$$
\frac{S_k}{1 - \alpha_k} \leq S_{k-1} + \delta_k + \mathbb{E}[\Delta_k(z_{k-1})],
\tag{21}
$$

For variant Algorithm 1, we have $\Delta_k(z_{k-1}) = 0$ since $w_k = x_k^\star$, and we obtain the following relation by unrolling the recursion:

$$
S_k \leq A_k \left( S_0 + \sum_{j=1}^{k} \frac{\delta_j}{A_{j-1}} \right) \qquad \text{with} \qquad A_j = \prod_{i=1}^{j}(1 - \alpha_i).
\tag{22}
$$

**Specialization to $\mu > 0$.** When $\mu > 0$, we have $\alpha_0 = \sqrt{q}$ and

$$
\begin{aligned}
S_0 &= (1 - \sqrt{q})(F(x_0) - F^\star) + \frac{\kappa \sqrt{q}(\sqrt{q} - q)}{2(1 - q)} \|x_0 - x^\star\|^2 \\
&= (1 - \sqrt{q})(F(x_0) - F^\star) + \frac{(\kappa + \mu)\sqrt{q}(\sqrt{q} - q)}{2} \|x_0 - x^\star\|^2 \\
&= (1 - \sqrt{q})(F(x_0) - F^\star) + \frac{\mu(1 - \sqrt{q})}{2} \|x_0 - x^\star\|^2 \\
&\leq 2(1 - \sqrt{q})(F(x_0) - F^\star),
\end{aligned}
\tag{23}
$$

by using the strong convexity inequality $F(x_0) \geq F^\star + \frac{\mu}{2}\|x_0 - x^\star\|^2$. Then, noting that $\mathbb{E}[F(x_k) - F^\star] \leq \frac{S_k}{1 - \sqrt{q}}$ and $A_k = (1 - \sqrt{q})^k$ (Lemma 8), we immediately obtain the first part of (8) from (22).

**Specialization to $\mu = 0$.** When $\mu = 0$, we have $\alpha_0 = 1$ and $S_0 = \frac{\kappa}{2}\|x_0 - x^\star\|^2$. Then, according to Lemma 8 and (22), for $k \geq 1$,

$$
\mathbb{E}[F(x_k) - F^\star] \leq \frac{S_k}{1 - \alpha_k} \leq \frac{\kappa \|x_0 - x^\star\|^2}{2} A_{k-1} + \sum_{j=1}^{k} \frac{\delta_j A_{k-1}}{A_{j-1}},
\tag{24}
$$

and we obtain the second part of (8) noting that $A_{k-1} \leq \frac{4}{(k+1)^2}$ and that $A_{j-1} \geq \frac{2}{(j+1)^2}$. Then, Proposition 1 is proven.

**Proof of Proposition 4.** When $w_k = x_k$, we need to control the quantity $\Delta_k(z_{k-1})$. Consider any scalar $\theta_k$ in $(0, 1)$. Then,

$$
\begin{aligned}
\Delta_k(z_{k-1}) &= -\frac{\kappa + \mu}{2}\|x_k - x_k^\star\|^2 - (\kappa + \mu)\langle z_{k-1} - x_k, x_k - x_k^\star\rangle \\
&= -\frac{\kappa + \mu}{2}\|x_k - x_k^\star\|^2 - (\kappa + \mu)\alpha_{k-1}\langle x^\star - v_k, x_k - x_k^\star\rangle \\
&\leq -\frac{\kappa + \mu}{2}\|x_k - x_k^\star\|^2 + (\kappa + \mu)\alpha_{k-1}\|x^\star - v_k\|\|x_k - x_k^\star\| \\
&\leq \left(\frac{1}{\theta_k} - 1\right)\frac{\kappa + \mu}{2}\|x_k - x_k^\star\|^2 + \frac{\theta_k(\kappa + \mu)\alpha_{k-1}^2}{2}\|x^\star - v_k\|^2 \quad \text{(Young's inequality)} \\
&\leq \left(\frac{1}{\theta_k} - 1\right)(h_k(x_k) - h_k^\star) + \frac{\theta_k(\kappa + \mu)\alpha_{k-1}^2}{2}\|x^\star - v_k\|^2 \quad \text{(since } \theta_k \leq 1) \\
&\leq \left(\frac{1}{\theta_k} - 1\right)(h_k(x_k) - h_k^\star) + \frac{\theta_k(\kappa + \mu)(\alpha_k^2 - \alpha_k q)}{2(1 - \alpha_k)}\|x^\star - v_k\|^2 \\
&= \left(\frac{1}{\theta_k} - 1\right)(h_k(x_k) - h_k^\star) + \frac{\theta_k \kappa \alpha_k \eta_k}{2(1 - \alpha_k)}\|x^\star - v_k\|^2.
\end{aligned}
$$

Then, we take expectations and, noticing that the quadratic term involving $\|x^\star - v_k\|^2$ is smaller than $\theta_k S_k/(1 - \alpha_k)$ in expectation (from the definition of $S_k$ in (20)), we obtain

$$
\mathbb{E}[\Delta_k(z_{k-1})] \leq \left(\frac{1}{\theta_k} - 1\right)\varepsilon_k + \frac{\theta_k S_k}{1 - \alpha_k},
$$

and from (21),

$$
S_k \leq \frac{(1 - \alpha_k)}{(1 - \theta_k)}\left(S_{k-1} + \delta_k + \left(\frac{1}{\theta_k} - 1\right)\varepsilon_k\right).
$$

By unrolling the recursion, we obtain

$$
S_k \leq \frac{A_k}{\Theta_k}\left(S_0 + \sum_{j=1}^{k}\frac{\Theta_{j-1}}{A_{j-1}}\left(\delta_j - \varepsilon_j + \frac{\varepsilon_j}{\theta_j}\right)\right) \quad \text{with} \quad A_j = \prod_{i=1}^{j}(1 - \alpha_i) \quad \text{and} \quad \Theta_j = \prod_{i=1}^{j}(1 - \theta_i). \tag{25}
$$

**Specialization to $\mu > 0$.** When $\mu > 0$, we have $\alpha_k = \sqrt{q}$ for all $k \geq 0$. Then, we may choose $\theta_k = \frac{\sqrt{q}}{2}$; then, $1 - \sqrt{q} \leq \left(1 - \frac{\sqrt{q}}{2}\right)^2$ and $\frac{A_k}{\Theta_k} \leq \left(1 - \frac{\sqrt{q}}{2}\right)^k$ for all $k \geq 0$. By using the relation (23), we obtain

$$
\begin{aligned}
S_k &\leq 2\left(1 - \frac{\sqrt{q}}{2}\right)^k(1 - \sqrt{q})(F(x_0) - F^\star) + 2\sum_{j=1}^{k}\left(\frac{1 - \sqrt{q}}{1 - \frac{\sqrt{q}}{2}}\right)^{k-j+1}\left(\delta_j - \varepsilon_j + \frac{\varepsilon_j}{\sqrt{q}}\right) \\
&\leq (1 - \sqrt{q})\left(2\left(1 - \frac{\sqrt{q}}{2}\right)^k(F(x_0) - F^\star) + 4\sum_{j=1}^{k}\left(\frac{1 - \sqrt{q}}{1 - \frac{\sqrt{q}}{2}}\right)^{k-j}\left(\delta_j - \varepsilon_j + \frac{\varepsilon_j}{\sqrt{q}}\right)\right) \\
&\leq (1 - \sqrt{q})\left(2\left(1 - \frac{\sqrt{q}}{2}\right)^k(F(x_0) - F^\star) + 4\sum_{j=1}^{k}\left(1 - \frac{\sqrt{q}}{2}\right)^{k-j}\left(\delta_j - \varepsilon_j + \frac{\varepsilon_j}{\sqrt{q}}\right)\right),
\end{aligned}
$$

where the second inequality uses $\frac{1}{1 - \frac{\sqrt{q}}{2}} \leq 2$. Since $(1 - \sqrt{q})\mathbb{E}[F(x_k) - F^\star] \leq S_k$, we obtain the first part of Proposition (4).

**Specialization to $\mu = 0$.** When $\mu = 0$, we have $\alpha_0 = 1$ and $S_0 = \frac{\kappa}{2}\|x_0 - x^\star\|^2$. We may then choose $\theta_k = \frac{\gamma}{(k+1)^{1+\gamma}}$ for any $\gamma$ in $(0, 1]$, leading to $e^{-(1+\gamma)} \leq \Theta_k \leq 1$ for all $k \geq 0$ according to Lemma 9. Besides, according to the proof of Lemma 8, $\frac{2}{(k+2)^2} \leq A_k \leq \frac{4}{(k+2)^2}$ for all $k \geq 1$.

Then, from (25),

$$\mathbb{E}[F(x_k) - F^\star] \leq \frac{A_{k-1}}{\Theta_k} \frac{\kappa \|x_0 - x^\star\|^2}{2} + \sum_{j=1}^{k} \frac{A_{k-1}\Theta_{j-1}}{\Theta_k A_{j-1}} \left(\delta_j - \varepsilon_j + \frac{\varepsilon_j}{\gamma}(1+j)^{1+\gamma}\right)$$

$$\leq \frac{2e^{1+\gamma}}{(k+1)^2} \left(\kappa\|x_0 - x^\star\|^2 + \sum_{j=1}^{k}(j+1)^2(\delta_j - \varepsilon_j) + \frac{(j+1)^{3+\gamma}\varepsilon_j}{\gamma}\right),$$

which yields the second part of Proposition (4). $\qquad\square$

## C.2 Proof of Proposition 5

Assume that for $k \geq 2$, we have the relation

$$\mathbb{E}[h_{k-1}(x_{k-1}) - h_{k-1}^\star] \leq \varepsilon_{k-1}. \tag{26}$$

Then, we want to evaluate the quality of the initial point $x_{k-1}$ to minimize $h_k$.

$$h_k(x_{k-1}) - h_k^\star = h_{k-1}(x_{k-1}) + \frac{\kappa}{2}\|x_{k-1} - y_{k-1}\|^2 - \frac{\kappa}{2}\|x_{k-1} - y_{k-2}\|^2 - h_k^\star$$

$$= h_{k-1}(x_{k-1}) - h_{k-1}^\star + h_{k-1}^\star - h_k^\star + \frac{\kappa}{2}\|x_{k-1} - y_{k-1}\|^2 - \frac{\kappa}{2}\|x_{k-1} - y_{k-2}\|^2$$

$$= h_{k-1}(x_{k-1}) - h_{k-1}^\star + h_{k-1}^\star - h_k^\star - \kappa(x_{k-1} - y_{k-1})^\top(y_{k-1} - y_{k-2}) - \frac{\kappa}{2}\|y_{k-1} - y_{k-2}\|^2. \tag{27}$$

Then, we may use the fact that $h_k^\star$ can be interpreted as the Moreau-Yosida smoothing of the objective $f$, defined as $G(y) = \min_{x\in\mathbb{R}^p} F(x) + \frac{\kappa}{2}\|x - y\|^2$, which gives us immediately a few useful results, as noted in [34]. Indeed, we know that $G$ is $\kappa$-smooth with $\nabla G(y_{k-1}) = \kappa(y_{k-1} - x_k^\star)$ for all $k \geq 1$ and

$$h_{k-1}^\star = G(y_{k-2}) \leq G(y_{k-1}) + \nabla G(y_{k-1})^\top(y_{k-2} - y_{k-1}) + \frac{\kappa}{2}\|y_{k-1} - y_{k-2}\|^2$$

$$= h_k^\star + \kappa(y_{k-1} - x_k^\star)^\top(y_{k-2} - y_{k-1}) + \frac{\kappa}{2}\|y_{k-1} - y_{k-2}\|^2. \tag{28}$$

Then, combining (27) and (28),

$$h_k(x_{k-1}) - h_k^\star \leq h_{k-1}(x_{k-1}) - h_{k-1}^\star + \kappa(x_{k-1} - x_k^\star)^\top(y_{k-2} - y_{k-1}).$$

$$\leq h_{k-1}(x_{k-1}) - h_{k-1}^\star + \kappa(x_{k-1} - x_{k-1}^\star)^\top(y_{k-2} - y_{k-1}) + \kappa(x_{k-1}^\star - x_k^\star)^\top(y_{k-2} - y_{k-1})$$

$$\leq h_{k-1}(x_{k-1}) - h_{k-1}^\star + \kappa(x_{k-1} - x_{k-1}^\star)^\top(y_{k-2} - y_{k-1}) + \kappa\|y_{k-1} - y_{k-2}\|^2$$

$$\leq h_{k-1}(x_{k-1}) - h_{k-1}^\star + \frac{\kappa}{2}\|x_{k-1} - x_{k-1}^\star\|^2 + \frac{3\kappa}{2}\|y_{k-1} - y_{k-2}\|^2$$

$$\leq \frac{3}{2}(h_{k-1}(x_{k-1}) - h_{k-1}^\star) + \frac{3\kappa}{2}\|y_{k-1} - y_{k-2}\|^2,$$

where the third inequality uses the non-expansiveness of the proximal operator; the fourth inequality uses the inequality $a^\top b \leq \frac{\|a\|^2}{2} + \frac{\|b\|^2}{2}$ for vectors $a, b$, and the last inequality uses the strong convexity of $h_{k-1}$. Then, we may use the same upper-bound on $\|y_{k-1} - y_{k-2}\|$ as [33, Proposition 12], namely

$$\|y_{k-1} - y_{k-2}\|^2 \leq 36\max\left(\|x_{k-1} - x^\star\|^2, \|x_{k-2} - x^\star\|^2, \|x_{k-3} - x^\star\|^2\right),$$

where we define $x_{-1} = x_0$ if $k = 2$.

## C.3 Proof of Proposition 6

The proof is similar to the derivation described in Section B.3.

**Inner-loop complexity.** With the choice of $\delta_k$, we have that $\xi_{k-1} = O(\delta_{k-1}/\sqrt{q})$. Besides, since we enforce $\mathbb{E}[H_k(x_k) - H_k^\star] \leq \delta_k$ for all $k \geq 0$, the result of Proposition 5 can be applied and the discussion following the proposition still applies, such that the complexity for computing $x_k$ is indeed $\tilde{O}(\tau/\eta_k)$.

**Outer-loop complexity.** Then, according to Proposition 1, it is easy to show that $\mathbb{E}[F(x_k) - F^\star] \leq O((1 - \sqrt{q}/2)^k (F(x_0) - F^\star))/\sqrt{q}$ and thus it suffices to choose

$$K = O\left(\frac{1}{\sqrt{q}} \log\left(\frac{F(x_0) - F^\star}{\sqrt{q}\varepsilon}\right)\right)$$

iterations to guarantee $\mathbb{E}[F(x_K) - F^\star] \leq \varepsilon$.

**Global complexity.** We use the exact same derivations as in Section B.3 except that we use the fact that $\varepsilon = O(\varepsilon_K/\sqrt{q}) = O((1 - \sqrt{q}/3)^K (F(x_0) - F^\star)/\sqrt{q})$ instead of $\varepsilon = O(\varepsilon_K/q)$, which gives us the desired complexity.

# D  Methods $\mathcal{M}$ with Duality Gaps Based on Strongly-Convex Lower Bounds

In this section, we summarize a few results from [28] and introduce minor modifications to guarantee the condition (14). For solving a stochastic composite objectives such as (1), where $F$ is $\mu$-strongly convex, consider an algorithm $\mathcal{M}$ performing the following classical updates

$$z_t \leftarrow \text{Prox}_{\eta\psi}[z_{t-1} - \eta g_t] \quad \text{with} \quad \mathbb{E}[g_t|\mathcal{F}_{k-1}] = \nabla f(z_{t-1}),$$

where $\eta \leq 1/L$, and the variance of $g_t$ is upper-bounded by $\sigma_t^2$. Inspired by estimate sequences from [40], the authors of [28] build recursively a $\mu$-strongly convex quadratic function $d_t$ of the form

$$d_t(z) = d_t^\star + \frac{\mu}{2}\|z_t - z\|^2.$$

From the proof of Proposition 1 in [28], we then have

$$\mathbb{E}[d_t^\star] \geq (1 - \eta\mu)\mathbb{E}[d_{k-1}^\star] + \eta\mu\mathbb{E}[F(z_t)] - \eta^2\mu\sigma_t^2,$$

which leads to

$$F^\star - \mathbb{E}[d_t^\star] + \eta\mu(\mathbb{E}[F(z_t)] - F^\star) \leq (1 - \eta\mu)\mathbb{E}[F^\star - d_{k-1}^\star] + \eta^2\mu\sigma_t^2,$$

which is a minor modification of Proposition 1 in [28] that is better suited to our purpose.

**With constant variance.** Assume now that $\sigma_t = \sigma$ for all $k \geq 1$. Following the iterate averaging procedure used in Theorem 1 of [28], which produces an iterate $\hat{z}_t$, we obtain

$$\mathbb{E}[F(\hat{z}_t) - d_t^\star] \leq (1 - \eta\mu)^t (F(z_0) - d_0^\star) + \eta\sigma^2, \tag{29}$$

where $d_0^\star$ can be freely specified for the analysis: it is not used by the algorithm, but it influences $d_t^\star$ through the relation $\mathbb{E}[d_t(z)] \leq \Gamma_t d_0(z) + (1 - \Gamma_t)\mathbb{E}[F(z)]$ with $\Gamma_t = (1 - \mu\eta)^k$, see Eq. (11) in [28]. In contrast, Theorem 1 in [28] would give here

$$\mathbb{E}[F(\hat{z}_t) - F^\star + d_t(z^\star) - d_t^\star] \leq (1 - \eta\mu)^t (2(F(z_0) - F^\star)) + \eta\sigma^2, \tag{30}$$

where $z^\star$ is a minimizer of $F$, which is sufficient to guarantee (2) given that $d_t(z^\star) \geq d_t^\star$.

**Application to the minimization of $H_k$.** Let us now consider applying the method to an auxiliary function $H_k$ from (14) instead of $F$, with initialization $x_{k-1}$. After running $T$ iterations, define $h_k$ to be the corresponding function $d_T$ defined above and $x_k = \hat{z}_T$. $H_k$ is $(\kappa + \mu)$-strongly convex and thus $h_k$ is also $(\kappa + \mu)$-strongly convex such that $(\mathcal{H}_1)$ is satisfied. Let us now check possible choices for $d_0^\star$ to ensure $(\mathcal{H}_2)$. For $z_{k-1} = \alpha_{k-1}x^\star + (1 - \alpha_{k-1})x_{k-1}$, $\mathbb{E}[d_T(z_{k-1})] \leq \Gamma_T d_0(z_{k-1}) + (1 - \Gamma_T)H_k(z_{k-1})$ such that we simply need to choose $d_0^\star$ such that $\mathbb{E}[d_0(z_{k-1})] \leq \mathbb{E}[H_k(z_{k-1})]$. Then, choose

$$d_0^\star = H_k^\star - F(x_{k-1}) + F^\star, \tag{31}$$

and

$$
\begin{aligned}
d_0(z_{k-1}) &= d_0^\star + \frac{\kappa + \mu}{2}\|x_{k-1} - z_{k-1}\|^2 \\
&= d_0^\star + \frac{(\kappa + \mu)\alpha_{k-1}^2}{2}\|x_{k-1} - x^\star\|^2 \\
&= d_0^\star + \frac{\mu}{2}\|x_{k-1} - x^\star\|^2 \\
&\leq d_0^\star + F(x_{k-1}) - F^\star = H_k^\star \leq H_k(z_{k-1}),
\end{aligned}
$$

such that $(\mathcal{H}_2)$ is satisfied, and finally (29) becomes

$$\mathbb{E}[H_k(x_k) - h_k^\star] \leq (1 - \eta(\mu + \kappa))^T (H_k(x_{k-1}) - H_k^\star + F(x_{k-1}) - F^\star) + \eta\sigma^2,$$

which matches (14).

**Variance-reduction methods.** In [28], gradient estimators $g_t$ with variance reduction are studied, leading to variants of SAGA [13], MISO [35], and SVRG [53], which can deal with the stochastic finite-sum problem presented in Section 1. Then, the variance of $\sigma_t^2$ decreases (Proposition 2 in [28]).

Let us then consider again the guarantees of the method obtained when minimizing $F$ with $\frac{\mu}{L} \leq \frac{1}{5n}$. From Corollary 5 of [28], we have

$$\mathbb{E}[F(\hat{z}_t) - F^\star + d_t(z^\star) - d_t^\star] \leq 8 (1 - \mu\eta)^t (F(x_0) - F^\star) + 18\eta\sigma^2,$$

and (2) is satisfied. Consider now two cases at iteration $T$:

- if $\mathbb{E}[d_T(z^\star)] \geq F^\star$, then we have $\mathbb{E}[F(\hat{z}_T) - d_T^\star] \leq 8 (1 - \mu\eta)^T (F(x_0) - F^\star) + 18\eta\sigma^2$.
- otherwise, it is easy to modify Theorem 2 and Corollary 5 of [28] to obtain

$$\mathbb{E}[F(\hat{z}_T) - d_T^\star] \leq (1 - \mu\eta)^T (2(F(x_0) - F^\star) + 6(F^\star - d_0^\star)) + 18\eta\sigma^2, \tag{32}$$

**Application to the minimization of $H_k$.** Consider now applying the method for minimizing $H_k$, with the same choice of $d_0^\star$ as (31), which ensures $(\mathcal{H}_2)$, and same definitions as above for $x_k$ and $h_k$. Note that the conditions on $\mu$ and $L$ above are satisfied when $\kappa = \frac{L}{5n} - \mu$ under the condition $\frac{L}{5n} \geq \mu$. Then, we have from the previous results, after replacing $F$ by $H_k$ making the right subsitutions

$$\mathbb{E}[H_k(x_k) - h_k^\star] \leq (1 - (\mu + \kappa)\eta)^T (8(H_k(x_{k-1}) - H_k^\star) + 6(F(x_{k-1}) - F^\star)) + 18\eta\sigma^2,$$

and (14) is satisfied.

**Other schemes.** Whereas we have presented approaches were $d_t$ is quadratic, [28] also studies another class of algorithms where $d_t$ is composite (see Section 2.2 in [28]). The results we present in this paper can be extended to such cases, but for simplicity, we have focused on quadratic surrogates.

# E  Additional Experimental Material

**Computing resources.** The numerical evaluation was performed by using four nodes of a CPU cluster with 56 cores of Intel CPUs each. The full set of experiments presented in this paper (with 5 runs for each setup) takes approximately half a day.

**Making plots.** We run each experiment five times and average the outputs. We display plots on a logarithmic scale for the primal gap $F(x_k) - F^\star$ (with $F^\star$ estimated as the minimum value observed from all runs). Note that for SVRG, one iteration is considered to perform two epochs since it requires accessing the full dataset every $n$ iterations on average.

## E.1  Additional experiments.

**Acceleration with no noise, $\delta = 0$.** We start evaluating the acceleration approach when there is no noise. This is essentially evaluating the original Catalyst method [33] in a deterministic setup in order to obtain a baseline comparison when $\delta = 0$. The results are presented in Figures 2 and 3 for the logistic regression problem. As predicted by theory, acceleration is more important when conditioning is low (bottom curves).

**Stochastic acceleration with no noise, $\delta = 0.01$ and $\delta = 0.1$.** Then, we perform a similar experiments by adding noise and report the results in Figures 4, 5, 6, 7. In general, the stochastic Catalyst approach seems to perform on par with the accelerated SVRG approach of [28] and even better in one case.

**Evaluating the square hinge loss.** In Figure 8, we perform experiments using the square hinge loss, where the methods perform similarly as for the logistic regression case, despite the fact that the bounded noise assumption does not necessarily hold on the optimization domain for the square hinge loss.

**Evaluating ill-conditioned problems.** Finally, we study in Figure 10 how the methods behave when the problems are badly conditioned. There, acceleration seems to work on ckn-cifar, but fails on gene and alpha, suggestions that acceleration is difficult to achieve when the condition number is extremely low.

Figure 2: Accelerating SVRG-like methods for $\ell_2$-logistic regression with $\mu = 1/(10n)$ (top) and $\mu = 1/(100n)$ (bottom) for $\delta = 0$. All plots are on a logarithmic scale for the objective function value, and the $x$-axis denotes the number of epochs. The colored tubes around each curve denote a standard deviations across 5 runs. They do not look symmetric because of the logarithmic scale.

Figure 3: Same plots as in Figure 2 when comparing SVRG and SAGA, with no noise ($\delta = 0$) with $\mu = 1/(10n)$ (top) and $\mu = 1/(100n)$ (bottom) .

Figure 4: Same plots as in Figure 2 for $\delta = 0.01$ with $\mu = 1/(10n)$ (top) and $\mu = 1/(100n)$ (bottom).

Figure 5: Same plots as in Figure 3 for $\delta = 0.01$ with $\mu = 1/(10n)$ (top) and $\mu = 1/(100n)$ (bottom).

Figure 6: Same plots as in Figure 2 for $\delta = 0.1$ with $\mu = 1/(10n)$ (top) and $\mu = 1/(100n)$ (bottom).

Figure 7: Same plots as in Figure 3 for $\delta = 0.1$ with $\mu = 1/(10n)$ (top) and $\mu = 1/(100n)$ (bottom).

Figure 8: Accelerating SVRG-like methods when using the squared hinge loss instead of the logistic for $\delta = 0$ (top) and $\delta = 0.1$, both with $\mu = 1/(10n)$.

Figure 9: Same plots as in Figure 8 for SVRG and SAGA, with $\delta = 0$ (top) and $\delta = 0.1$ for $\mu = 1/(10n)$.

Figure 10: Illustration of potential numerical instabilities problems when the problem is very ill-conditioned. We use $\mu = 1/(1000n)$ with $\delta = 0$ for the logistic loss (top) and squared hinge (bottom).