[Reviews · NeurIPS 2019]

Reviewer 1



Summary: Catalyst (reference [34]) was a method proposed for taking some iterative algorithm M and producing a new algorithm M' that uses M as a subroutine but converges at an accelerated rate compared to M. That prior work applied to gradient-based methods for convex composite (smooth + non-smooth) problems and regularized empirical risk minimization problems. Catalyst (Algorithm 2 in [34]) uses a particular choice of surrogate function h_k. It appears that the main difference between Catalyst and the proposed "Generic Acceleration Framework with Inexact Minimization of hk" (Algorithm 2 in the current manuscript) is that h_k is no longer fixed and can be chosen from a more generic class of functions that satisfies certain properties. This enables the analysis and acceleration of algorithms that could not be treated with Catalyst, including Accelerated prox gradient method and Accelerated SGD with prox. This results in accelerated rate guarantees that have two terms: the first bounds convergence rates for the initialization-dominated regime, and the second bounds the convergence rate for the noise-dominated regime. Comments/Questions: 1) The manuscript is clear and well-written, and the literature search is comprehensive. It was a pleasure to read! I did not check the proofs in the appendix in detail, but the results make sense and the parts I did check appeared to be correct. 2) The treatment of different algorithms (lines 200-225) requires different choices of the surrogate function h_k. The authors provide no insight or discussion regarding the choice of h_k. How were these surrogate functions obtained? What happens if a different h_k is chosen? 3) The plots in Section 5 have no legends and are unlabeled. So it's not clear what the different curves represent. The tick labels are cut off on the top row of plots as well. The legends are present in the Appendix versions, but they should be reproduced in the main text as well. 4) The authors comment that the acceleration method may not perform well for very ill-conditioned problems, and show supporting empirical evidence. This is worrisome because there is no explanation of why this lack of performance occurs. Can this breakdown be predicted from theory?

Reviewer 2



I've read the other reviews and the rebuttal. I still believe that the contribution of the work is significant and keep my score - this is a good paper. I hope that all the minor issues will be fixed in the final version. ----------------------------------------------------- Originality. The proposed method and its analysis are based on previously known generic accelerated framework Catalyst, and generalized it to stochastic setting. Quality. The paper is basically well-written. It contains a number of theoretical results (with proofs in the supplementary) as well as experimental section. The part with contributions is a bit repetitive sometimes. Clarity. The results are well described on a conceptual level, but I would prefer to have more precise theoretical statements (with a definition of all symbols, procedures, etc.), especially in Section 3 (about multi-stage schemes) and in Section 4 (about applicational cases). Through out the paper, there is a plenty of notation which is used without preliminary definitions (or hidden inside the text). And it probably needs some clarifications (are norms Euclidean? Definitions of mu, L, sigma, ...) Significance. The results seem significant to me as a part of comprehensive study of stochastic convex optimization methods.

Reviewer 3



The paper extends Catalyst approach for general stochastic optimization. The paper is very well written and easy to follow. The authors are aware of relevant literature and are citing appropriate papers. Although I did not go through all proofs, the obtained results are believable and reasonable. Experiments -- it is not clear how the dropout is exactly used -- do you see the objective as one layer NN and use dropout on top of it just in order to have a stochastic objective? Is there some evidence that dropout helps for such shallow models in terms of generalization? Personally, I would appreciate explicitly stating what is h_k/H_k in the considered special cases (this partially overlaps with the latter point) somewhere in the appendix. Now, the reader has to guess for example what is the setting in the table from sec. 4.2 and why SGD is faster than SVRG/SAGA/MISO (I guess the reason is that the variance reduced methods assume general each f_i to be an expectation with the same bound on variance as the SGD objective). Lastly, the legend is missing in Figure 1. All above-mentioned issues are minor. The only real concern I have is in terms of the significance of the contribution, which is why I find the paper to be borderline. I do not find the results to be very hard to get but at the same time, the work is filling blank spots in the optimization world. In particular, the most important thing the method can do is to accelerate variance reduced methods with noise; however, as the authors pointed out, it is not the first method to do so (SVRG is already done, and one does not need to pay extra log factor there). Further, it is a community consensus that the Catalyst approach does not perform as well in practice as directly accelerated methods -- thus I do not see this work to have a significance for the practitioners (experiments (especially in the appendix) somewhat confirm slight superiority of accelerated SVRG). However, that is the price for the great versatility of Catalyst. Similarly, I do not believe the method will have a significant impact in the future nor it presents surprising results. On the other hand, the paper presents, to best of my knowledge, the most universal framework for the acceleration. I would strongly vote to accept the paper to the slightly lesser venue, however, at this moment I am rather indifferent. ========================= UPDATE ========================= All of my minor issues were addressed in the rebuttal. I am still not completely convinced about the practicality aspect and the high significance of the contribution, although that might be admittedly subjective. For this reason, I have decided to keep my score as it is (6). However, I believe that some discussion about the practicality of catalyst similar to what presented in the rebuttal would improve the paper -- it is both good to fix community's misconceptions (shall the authors believe it is a misconcepton) and it also makes the better motivation for the paper's results.

Reviewer 4



==================================== Strength/weakness/questions/suggestions: ==================================== 1- In line 56, it is mentioned that “we extend the Catalyst approach [34] to stochastic problems,” as the reviewer is aware of Catalyst method, [34] talks about the stochastic problems too, however, not exactly the same as this paper. It would be more clear if the authors clarify more about it. 2- In line 72, eq. (3), the right term, $ \sigma^2 / \mu \eps, can be very big when $\eps$ or $\mu$ (similar to the selected $ \mu $ in the numerical section in this paper) is small or when $\sigma$ is big. It means that the oracle complexity can be huge. 3- In line 126, if $ \tao = 1/(2 t^’) $, then $ \tao $ will decrease, or equivalently, the slower convergence rate. It is suggested that the authors clarify more about it. 4- In Section 4, it is not mentioned how $ \kappa $ will be chosen. In [34], $ \kappa $ is calculated based on a small optimization problem. It is suggested that the authors clarify more about how to select $ \kappa $. 5- In line 139, it is mentioned that “Note that the conditions on hk bear similarities with estimate sequences introduced by Nesterov [41]” however, this is not a good analogy. In [41], the assumptions are valid for deterministic case and it is not the same to say that those conditions are valid for stochastic case too. 6- The assumptions, in lines 135- 138 are strong (in algorithmic view) though the authors talked about some examples in page 4 and 5. 7- In line 154, it is mentioned that “we recover the accelerated rates of of [21]”, however, that conclusion is based on exact minimization of $ h_k$, therefore, it is not true to claim the recovery the accelerated rates of [21]. 8- In Algorithm 2, in step 4, why is the assumption $H_3$ still there? 9- In line 185, what is the relation between $f$ and $\eps_k$? According to assumption $H_4$, $\eps_k$ is related to $h_k$. 10- In line 197, it is mentioned that “when $\sigma =0 $...” however, $\sigma =0 $ means zero variance for stochastic case which is an unreal case. 11- The Figure 1 should have legend. 12- The x-axis in the first row of Figure 1 is not clear (the reviewer understands the number of pages are limited, however, the figures and text should be clear) 13- In page 9, the references [9] and [10] are the same. ==================================== Typos: ==================================== 1- Line 154, “rates of of [21]”, one of the “of” should be erased. 2- Line 246, “Indeed, we may notice that Therefore, even though the”, this sentence is not complete. 3- Line 509 and before line 510, it is needed to put a full stop 4- Line 515, in the second inequality no comma is needed ==================================== Comments about code: ==================================== 1- That would be better if a “Readme” file was provided how to run the code 2- The codes are well structured specially the file “svm.h” which the solvers are available there, however, it would be suggested to write more comments that the reviewer could follow it easier

[Author Response · NeurIPS 2019]

We thank the reviewers for their insightful comments and respond to their concerns and questions below.

**Choice of $h_k$ (R1).**  We consider $h_k$ to be either Eq. 4 (proximal point), Eq. 9 (proximal gradient), or Eq. 10
(prox SGD). In principle, it is possible to design other surrogates, which would lead to new algorithms coming with
convergence guarantees given by Prop. 1 and 4, but the three previous examples already cover important cases.

**Plots in Section 5 (R1,R2,R4,R5).**  We thank the reviewers for noting a few problems with Fig. 1. As noted by R1,
the legends are available in the appendix, and we will add them in the main text, if the paper is accepted.

**Acceleration for ill-conditioned problems (R1).**  In the appendix, we conduct an experiment with an extremely
high condition number (much higher than what is traditionally used for these problems). This is a case where the
sublinear convergence rates for convex optimization ($\mu = 0$) are typically better than the linear rates for strongly convex
optimization ($\mu > 0$), as long as the number of iterations is smaller than $L/\mu$. In such a situation, a better strategy
would have been not to assume the objective to be strongly convex at all. We will clarify this.

**Clarity (R2).**  We agree that the clarity of our paper is subject to improvement and we thank the reviewer for his
suggestions, which we will take into account, if the paper is accepted. Note that the norms are indeed Euclidean.

**Code (R2,R5).**  All algorithms are implemented in C++ in the file utils/svm.h and mex files for Matlab are given in
the folder mex/* Compiling requires (i) installing the Intel C++ compiler 2017, (ii) using Matlab 2016a, (iii) setting up
the path to the Intel libraries in build.m, (iv) typing "build" in matlab. We admit that this procedure is not friendly and
we will do our best to make it simpler in the future. Reproducing the results also requires accessing the datasets, which
we will provide on an external website after publication since they were too large to be uploaded in the supplementary.

**ckn-cifar (R2).**  We consider images encoded by unsupervised CKNs, and use our algorithms for the classification
layer, which is convex. Addressing non-convex problems would be very interesting, but beyond the scope of our paper.

**Dropout (R4).**  We use DropOut as in "Wager et al. Altitude Training: Strong Bounds for Single-Layer Dropout.
NIPS 2014.", which displays moderate gains on text classification tasks. However, the choice of DropOut in our
paper (vs. more realistic data augmentation strategies for images, see [7]) is mainly motivated by the need of a simple
optimization benchmark illustrating stochastic finite-sum problems, where the amount of perturbation is easy to control.

**Clarity (R4).**  We thank the reviewer for his suggestion about adding a table for $h_k/H_k$, which we will do.

**Significance (R4).**  We agree that direct acceleration methods are appealing. However, we also believe that the
practical benefits of Catalyst are often underestimated. In Alg. 2 of [34], Catalyst is presented with three variants a), b)
and c). Variants a) and b) require estimating optimality gaps for the sub-problems, which requires a lot of care (and
pain) when implementing the method. Variant c) uses a fixed budget in the inner loops and is trivial to implement, e.g.,
see the file svm.h in the code we provide. Because variant c) came later, the community has focused on a) and b), often
refering to Catalyst as being theoretically appealing but not practical. Yet, experiments conducted in [34] show that the
simple strategy c), which we follow in our paper, is more effective in general than a) and b) in practice.

Finally, we believe that a universal acceleration framework for stochastic optimization is also a significant contribution
from a conceptual/methodological point of view, but this is of course a subjective statement. It also provides a new point
of view on stochastic proximal point methods, which have recently gained some attention in machine learning, see [3].

**10 questions from R5.**  We thank the reviewer for his questions, which will lead to clarifications in the paper.
1. We will precise that [34] addresses only deterministic objectives (which may be finite sums).
2. We agree that a large variance combined with small $\mu$ leads to a large complexity, but this unfortunate combination
would affect any stochastic optimization method since $O(\sigma^2/\mu\varepsilon)$ is asymptotically optimal. The goal of variance-
reduction for the stochastic finite-sum problem is then precisely to reduce $\sigma^2$.
3. We will clarify that $t' = \lceil (2D/\mu)^{1/d} \rceil$ and $\tau = 1/2t'$ are fixed quantities, thus $\tau$ does not decrease. When writing
$t = st'$, we simply mean that we consider (2) after $s$ restarts, corresponding to $st'$ iterations of $\mathcal{M}$.
4. $\kappa$ is chosen as in [34] for the deterministic case $\sigma = 0$, leading to the right near-optimal complexity, even for $\sigma > 0$.
5.-6. $\mathcal{H}_2$ and $\mathcal{H}_3$ may look like strong assumptions, but they resemble classical ones used in the definition of estimate
sequences by Nesterov (from the deterministic world), see Eq. (2.2.1) and (2.2.2) of [41]. We will clarify the analogy.
7. In this paragraph, we recover the results of Sec. 2 of [21] only, which treats the case with exact minimization,
whereas we recover later the case with inexact minimization in Prop. 4.
8. When using the Catalyst surrogate, $\mathcal{H}_4$ implies $\mathcal{H}_3$ with $\delta_k = \varepsilon_k$. However, keeping both assumptions allows us to
address more exotic cases (line 220 to 225). For clarity, it may however be better to treat this case only in the appendix.
9. Given any sequence $(\varepsilon_k)_{k\geq 0}$ and surrogate satisfying $\mathcal{H}_4$, the relation between $F$ and $(\varepsilon_k)_{k\geq 0}$ is given in Prop. 4.
10. The goal of lines 197-200 is to illustrate our results when the objective is deterministic (which is what we mean by
$\sigma = 0$). In such a case, we show that we achieve acceleration and recover the Catalyst method of [34].
We also thank the reviewer for correcting some typos and noting the problem with references [9] and [10].

[Meta-Review · NeurIPS 2019]

Thank you for the nice paper. We have agreed to accept it for NeurIPS, however, please read carefully the reviews one more time and incorporate suggestions from reviewers into your final submission. Thanks